# CONFIDENCE-AWARE REWARD OPTIMIZATION FOR FINE-TUNING TEXT-TO-IMAGE MODELS

**Kyuyoung Kim**[1]    **Jongheon Jeong**[2*]   **Minyong An**[3]
**Mohammad Ghavamzadeh**[4]    **Krishnamurthy Dvijotham**[5]    **Jinwoo Shin**[1]    **Kimin Lee**[1]
[1]KAIST    [2]Korea University    [3]Yonsei University    [4]Amazon    [5]Google DeepMind

## ABSTRACT

Fine-tuning text-to-image models with reward functions trained on human feedback data has proven effective for aligning model behavior with human intent. However, excessive optimization with such reward models, which serve as mere proxy objectives, can compromise the performance of fine-tuned models, a phenomenon known as reward overoptimization. To investigate this issue in depth, we introduce the Text-Image Alignment Assessment (TIA2) benchmark, which comprises a diverse collection of text prompts, images, and human annotations. Our evaluation of several state-of-the-art reward models on this benchmark reveals their frequent misalignment with human assessment. We empirically demonstrate that overoptimization occurs notably when a poorly aligned reward model is used as the fine-tuning objective. To address this, we propose TextNorm, a simple method that enhances alignment based on a measure of reward model confidence estimated across a set of semantically contrastive text prompts. We demonstrate that incorporating the confidence-calibrated rewards in fine-tuning effectively reduces overoptimization, resulting in twice as many wins in human evaluation for text-image alignment compared against the baseline reward models.

## 1 INTRODUCTION

Large-scale text-to-image models have been successful in generating high-quality and creative images given text prompts as input (Saharia et al., 2022; Rombach et al., 2022; Yu et al., 2022). However, current models have several weaknesses (Hu et al., 2023), including limited ability in compositional generation (Feng et al., 2023), inaccurate text rendering (Liu et al., 2022), and difficulty with spatial understanding (Gokhale et al., 2022). Moreover, large-scale datasets used to train state-of-the-art text-to-image models often contain malicious content (Schuhmann et al., 2021) and undesirable biases (Fan et al., 2023) that models can potentially learn from during training.

*Learning from human feedback* has emerged as a powerful method for addressing these limitations and aligning text-to-image models with human intent (Fan et al., 2023; Lee et al., 2023; Xu et al., 2023). This method involves learning a reward function that approximates human objective (i.e., the true objective) from human feedback data, followed by optimizing the model against the learned reward to enhance alignment. However, optimizing too much against proxy objectives can hinder the true objective, a phenomenon commonly known as *reward overoptimization* (Gao et al., 2023).

In this work, we investigate the issue of overoptimization in text-to-image generation, evaluating several state-of-the-art reward models. To facilitate our evaluation, we introduce the Text-Image Alignment Assessment[1] (TIA2) benchmark, a diverse compilation of text prompts, images, and human annotations. Our findings indicate that reward models fine-tuned on human feedback data, such as ImageReward (Xu et al., 2023) and PickScore (Kirstain et al., 2023), exhibit stronger correlations with human assessments compared to pre-trained models like CLIP (Radford et al., 2021). Nevertheless, all of the models struggle to fully capture human preferences. As Figure 1 demonstrates, excessive optimization can compromise both text-image alignment and image fidelity. We show that this is particularly the case when the reward signal is poorly aligned with human judgment.

---

[*]Work done at KAIST.
[1]The benchmark data is available at https://github.com/kykim0/TextNorm.

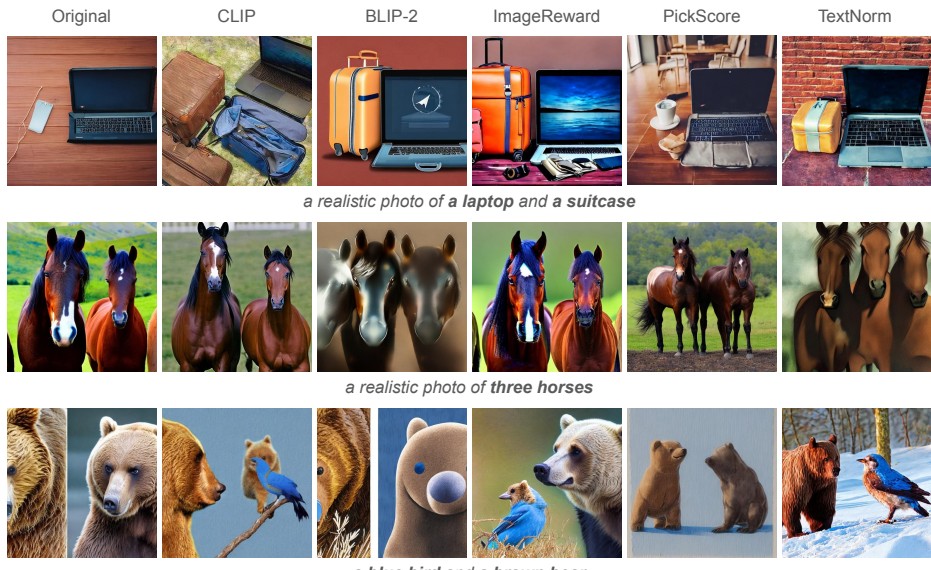

Figure 1: Images generated using Stable Diffusion v2.1 (Rombach et al., 2022) fined-tuned with CLIP (Radford et al., 2021), BLIP-2 (Li et al., 2023), ImageReward (IR; Xu et al. 2023), and PickScore (PS; Kirstain et al. 2023). Both text-image alignment and image fidelity exhibit degradation when subjected to excessive optimization. Our proposed method (TextNorm) demonstrates its robustness against overoptimization, as illustrated in the last column.

To alleviate overoptimization, we propose *textual normalization* (TextNorm), a simple method to enhance the alignment of reward models by calibrating rewards based on a measure of model confidence. Specifically, we consider a set of semantically contrastive prompts and adjust the reward conditioned on the input prompt relative to those conditioned on the contrastive prompts. The key idea is to leverage the relative comparison of the rewards as a measure of model confidence to calibrate rewards. In constructing the set of contrastive prompts, we show that both simple rule-based approaches and leveraging large language models (LLMs) such as ChatGPT (OpenAI, 2022) can be effective. We also propose and demonstrate that ensemble methods, which combine multiple reward models, can be used to achieve further improvement. Our experimental results demonstrate that TextNorm significantly enhances alignment with human judgment on the TIA2 benchmark. This improvement renders the fine-tuning of text-to-image models more robust against overoptimization, a conclusion supported by human evaluations.

Our main contributions are as follows:

- We introduce a benchmark for evaluating reward models in text-to-image generation, providing key insights into the alignment of several state-of-the-art models with human judgment.

- We empirically demonstrate the adverse effects of excessive optimization against learned reward models. Importantly, we show that overoptimization is conceivable, even for reward models trained on extensive human preference data.

- We propose TextNorm, a simple method for enhancing alignment by calibrating rewards based on a measure of model confidence. Through extensive experiments, we demonstrate that our approach could substantially mitigate overoptimization.

## 2    RELATED WORK

**Text-to-image generation.** Diffusion model (Sohl-Dickstein et al., 2015; Ho et al., 2020) is a family of generative models that have achieved state-of-the-art results across various domains, including image synthesis (Dhariwal & Nichol, 2021), 3D synthesis (Poole et al., 2023), and robotics (Chi et al., 2023). Different conditioning mechanisms guide the diffusion process allowing, for instance,

generation of images from textual descriptions. This has led to the development of many of the recent state-of-the-art text-to-image models (Rombach et al., 2022; Ramesh et al., 2022; Saharia et al., 2022). However, generating images that fully respect the input prompt remains a challenge.

**Evaluating text-to-image generation.** Evaluation of text-to-image models often requires considerable human effort, prompting a rising interest in the development of automatic evaluation methods. Vision-language models (VLMs), such as CLIP (Radford et al., 2021) and BLIP (Li et al., 2023), measure text-image alignment using the cosine similarity between their embeddings. Methods such as DALL-Eval (Cho et al., 2023) and VISOR (Gokhale et al., 2022) utilize object detection to determine whether the objects are present in the correct quantities and locations. ImageReward (Xu et al., 2023) and PickScore (Kirstain et al., 2023) fine-tune VLMs on a large set of human preference data to directly predict preference. In this work, we examine multiple of these models and propose a method for improving their alignment with human judgment.

**Reward overoptimization.** Training a reward model on human feedback data and fine-tuning LLMs on this reward signal has proven effective in aligning the models with human objectives (Stiennon et al., 2020; Ouyang et al., 2022). Similar methods have been applied to improve text-to-image models (Lee et al., 2023; Xu et al., 2023; Fan et al., 2023), enabling the models to generate more human-preferred images. However, excessive optimization against a reward model, which is an imperfect proxy, can degrade model quality (Gao et al., 2023; Black et al., 2023). In this study, we present a comprehensive empirical demonstration of overoptimization in text-to-image generation and show that employing better calibrated rewards can effectively alleviate the issue.

## 3 TIA2: A BENCHMARK FOR TEXT-IMAGE ALIGNMENT ASSESSMENT

### 3.1 DATA COLLECTION AND STATISTICS

We collect 100 text prompts sampled from diverse sources, including DrawBench (Saharia et al., 2022), PartiPrompt (Yu et al., 2022), ImageRewardDB (Xu et al., 2023), and Pick-a-Pic (Kirstain et al., 2023), to construct the *comprehensive* set. We additionally create synthetic prompts describing quantities of objects or combinations of two distinct objects to form the *counting* and *composition* sets. For the synthetic prompts, we utilize a total of 25 object classes, comprising 10 from CIFAR-10 (Krizhevsky et al., 2009) and 15 from MS-COCO (Lin et al., 2014). The counting set contains prompts describing the quantity of objects from 1 to 6 for each object class, such as "three deer," while the composition set comprises prompts describing combinations of two distinct object classes, such as "a cat and a dog." For each prompt, we generate 50 images using Stable Diffusion v2.1 (Rombach et al., 2022) to include in the benchmark. Table 1 provides basic statistics on the benchmark. More details on the dataset can be found in Appendix B.

To evaluate text-image alignment, we gather binary feedback (*good* or *bad*) from three human annotators for each text-image pair. We consolidate annotators' responses by assigning a good label if at least two out of three are positive; otherwise, we assign a bad label. Figure 2 illustrates two sets of images, highlighting instances where the reward models fail to fully capture human assessment.

### 3.2 BENCHMARKING REWARD MODELS

**Problem setup.** Our benchmark presents a dataset $\mathcal{D} := \{(x_i, y_i, z_i)\}_{i=1}^{n}$ comprising triplets $(x, y, z)$ of a text prompt $x \in \mathcal{X}$, an image $y \in \mathcal{Y}$, and a binary human label $z \in \{0, 1\}$ indicating whether $x$ and $y$ are semantically consistent ("1") or not ("0"). The goal of reward modeling for predicting text-image semantic consistency is then to derive a function $r(x, y) \in \mathbb{R}$ based on which we can infer the label $z$ for the prompt $x$ and the image $y$. In this view, we can frame reward modeling as a binary classification task and assess reward models based on their performance as binary classifiers of human labels. For instance, rewards computed using $r$ can be converted to binary predictions based on a chosen threshold, and then compared to the human labels.

**Evaluation metrics.** We employ standard metrics for binary classification, including the Area Under the Receiver Operating Characteristic (AUROC) and the Area Under the Precision-Recall Curve (AUPRC), to evaluate the ability of reward models as classifiers in distinguishing between semantically consistent and inconsistent samples. These threshold-independent metrics are used to assess the overall alignment with human labels. When reward models are used as optimization objective,

Table 1: Statistics on the TIA2 benchmark. The benchmark consists of a total of 550 text prompts, 27,500 images, and a set of three human annotations for every text-image pair. See Appendix B for more details on the benchmark.

| Category | Counts | | Labels (%) | |
|---|---|---|---|---|
| | Texts | Images | Good | Bad |
| Comprehensive | 100 | 5,000 | 47.22 | 52.78 |
| Counting | 150 | 7,500 | 43.27 | 56.73 |
| Composition | 300 | 15,000 | 38.97 | 61.03 |
| Total | 550 | 27,500 | 41.64 | 58.36 |

| **Prompt**: an apple and a deer | |
|---|---|
| Human label | |
| **Good** | Bad |
| CLIP | |
| **0.379** | 0.360 |
| BLIP-2 | |
| 0.387 | **0.458** |
| ImageReward | |
| 0.800 | **1.453** |
| PickScore | |
| **0.227** | 0.208 |

| **Prompt**: a book and a teddy bear | |
|---|---|
| Human label | |
| **Good** | Bad |
| CLIP | |
| 0.324 | **0.352** |
| BLIP-2 | |
| 0.386 | **0.417** |
| ImageReward | |
| 1.288 | **1.459** |
| PickScore | |
| 0.223 | **0.225** |

Figure 2: Sample images for which the reward models do not fully agree with human labels.

ensuring that the high-scoring samples align closely with human assessment becomes particularly important. To gauge this alignment, we use average precision (AP) at $k \in \{5, 10, 25\}$ to evaluate how well the rewards match the human labels for the top-$k$ samples. Lastly, we consider rank correlation statistics, namely Spearman's $\rho$ and Kendall's $\tau$, as additional metrics to assess the similarity between the rewards and human labels. For a more fine-grained comparison, we aggregate the three individual human labels for each example. We convert a "good" label to 1, a "bad" label to 0, and an "inconclusive" label to 0.5. The score is then computed as the mean of these three numbers, and rank correlations are calculated between the rewards and these aggregate scores. We compute the metrics for each prompt using the 50 samples provided in the benchmark.

## 4 CONFIDENCE-CALIBRATED REWARD DESIGN

In this section, we explore methods that enhance the alignment of reward models based on a measure of model confidence. We quantitatively assess a range of reward models on our benchmark, showing that even those fine-tuned on human feedback data often inadequately capture human preferences. Subsequently, we demonstrate the effectiveness of our method in comparison.

### 4.1 TEXTUAL NORMALIZATION

Given a reward model $r(x_0, y) \in \mathbb{R}$ that measures text-image semantic consistency between a text prompt $x_0$ and an image $y$, we propose a simple method that leverages a measure of reward model confidence for enhancing its alignment. Specifically, we consider a set of alternative prompts $X$, and normalize the reward $r(x_0, y)$ using the softmax function based on the rewards conditioned on these prompts. The idea is to view the relative comparison of the rewards as a measure of model confidence in $r(x_0, y)$ and calibrate the reward accordingly. For instance, if the rewards conditioned on the alternative prompts are relatively close to $r(x_0, y)$, it suggests that the model is less confident in $r(x_0, y)$, prompting a more significant adjustment in the reward. This softmax-based approach, while simple, can be effective in estimating model confidence (Hendrycks & Gimpel, 2017).

With access to all semantically distinct prompts, we could compute this model confidence precisely. However, computing the softmax function over this prompt set would involve an infinite sum. Instead, we approximate this sum by considering a finite set $X$ of prompts each of which shares syntactic similarity with $x_0$ while being semantically distinct. The rationale behind this approach is based on the hypothesis that prompts that differ from $x_0$ in both syntax and semantics are unlikely to yield high reward values $r(x, y)$, thus making negligible contributions to the softmax score. We empirically evaluate this hypothesis and confirm that it holds in practice. Using the set of prompts $X = \{x_j\}_{j=1}^m$, we define the following score for the reward model $r$:

$$r^X(x_0, y) := \frac{\exp\left(\frac{1}{\tau} \cdot r(x_0, y)\right)}{\exp\left(\frac{1}{\tau} \cdot r(x_0, y)\right) + \sum_{i=1}^m \exp\left(\frac{1}{\tau} \cdot r(x_i, y)\right)}, \quad (1)$$

where $\tau > 0$ is a temperature scale.

To illustrate the types of prompts in $X$, consider the text prompt $x_0 = $ *"a photo of two dogs"*. We could include prompts such as *"a photo of three dogs"* and *"a photo of three cats"* that are

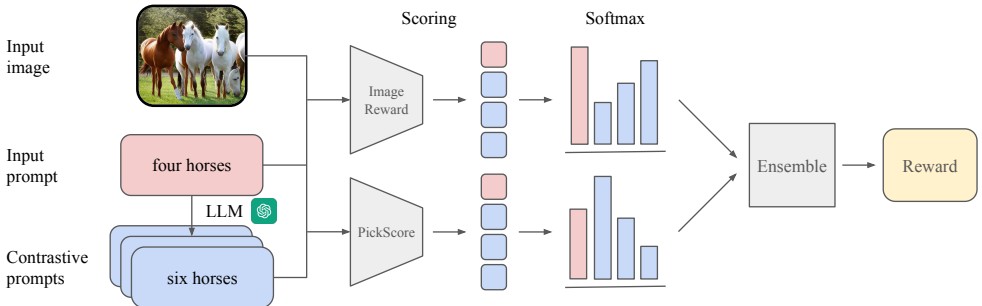

Figure 3: TextNorm normalizes rewards over a set of contrastive prompts generated using an LLM. Combining an ensemble of normalized rewards of multiple models can further enhance alignment.

syntactically similar to but semantically different from $x_0$. In contrast, prompts such as *"describe a colorful abstract painting"* would not be as useful to be part of $X$.

**Prompt set synthesis via language models.** An important step in implementing TextNorm involves creating contrastive prompts over which to normalize rewards. For simple input prompts, a rule-based approach can often be effective. For instance, when the input prompt describes a quantity of an object, prompts that describe various other quantities of the same object can be used. For more linguistically complex input prompts, we can leverage LLMs to generate contrastive prompts, providing appropriate few-shot examples to guide the generation:

*"Create captions that are different from the original input used for the text-to-image generation model, referencing the provided failure cases ... [few-shot examples] ... [input prompt] ..."*

When composing the LLM prompt, we analyze common failure scenarios of text-to-image models for the type of the input $x_0$ and present their textual descriptions as few-shot examples. For instance, we observed that models often generate images containing incorrect quantities or types of objects compared to those described in $x_0$. Incorporating these prompts into $X$ results in better calibrated rewards, particularly when the image is indeed inconsistent with the input prompt. Also, depending on the nature of $x_0$, we include few-shot examples considering other relevant aspects such as spatial relationship, color, and material. See Appendix D for further details on the use of LLMs.

## 4.2 REWARD MODEL ENSEMBLE

On our benchmark, we found PickScore to be overall better aligned with human judgment than other baselines (see Section 4.3 for a more comprehensive evaluation). However, we noted instances where competitive baselines such as ImageReward showed stronger alignment, as shown in the examples in Appendix B.4. Motivated by the observation, we propose combining an ensemble of reward models to further enhance alignment. Given a set of reward models $\{r_1, \ldots, r_k\}$, we first apply TextNorm to derive corresponding rewards $\{r_1^X, \ldots, r_k^X\}$ normalized over the set of prompts $X$. This ensemble of normalized rewards can then be combined in several natural ways. See Figure 3 for an illustration of an overview of the method.

**Mean ensemble.** A method as simple as averaging can be effective, particularly when the reward models within the ensemble are of comparable quality.

$$r_\mu^X(x_0, y) := \frac{1}{k} \sum_i^k r_i^X(x_0, y),$$  (2)

where $x_0$ is the prompt, and $y$ denotes the image.

**Uncertainty-regularized ensemble.** If the reward models in the ensemble disagree significantly, a penalty based on the variance of the normalized rewards can act as a form of regularization. That is,

$$r_\lambda^X(x_0, y) := r_\mu^X(x_0, y) - \lambda \cdot \frac{1}{k} \sum_i^k \left( r_i^X(x_0, y) - r_\mu^X(x_0, y) \right)^2,$$  (3)

where $x_0$ is the prompt, $y$ denotes the image, and $\lambda$ is the weight of the variance penalty.

Table 2: Alignment evaluation on each of the comprehensive (C100), counting (C150), and composition (C300) sets. Prompts for which all human labels are identical are excluded.

|      | Reward | AUROC | AUPRC | AP@5 | AP@10 | AP@25 | Spearman | Kendall |
|------|--------|-------|-------|------|-------|-------|----------|---------|
| C100 | CLIP | 0.671 | 0.611 | 0.674 | 0.667 | 0.635 | 0.235 | 0.187 |
|      | BLIP-2 | 0.683 | 0.603 | 0.648 | 0.642 | 0.616 | 0.233 | 0.185 |
|      | ImageReward | 0.761 | 0.674 | 0.755 | 0.740 | 0.708 | 0.361 | 0.289 |
|      | PickScore | 0.747 | 0.675 | 0.779 | 0.763 | 0.706 | 0.325 | 0.258 |
|      | **TextNorm** | **0.788** | **0.727** | **0.841** | **0.820** | **0.765** | **0.372** | **0.298** |
| C150 | CLIP | 0.595 | 0.522 | 0.622 | 0.600 | 0.550 | 0.136 | 0.107 |
|      | BLIP-2 | 0.565 | 0.499 | 0.570 | 0.556 | 0.519 | 0.072 | 0.056 |
|      | ImageReward | 0.657 | 0.543 | 0.611 | 0.605 | 0.570 | 0.208 | 0.166 |
|      | PickScore | 0.731 | 0.614 | 0.711 | 0.702 | 0.650 | 0.340 | 0.273 |
|      | **TextNorm** | **0.807** | **0.719** | **0.877** | **0.841** | **0.763** | **0.443** | **0.356** |
| C300 | CLIP | 0.717 | 0.607 | 0.737 | 0.699 | 0.641 | 0.431 | 0.333 |
|      | BLIP-2 | 0.613 | 0.506 | 0.579 | 0.554 | 0.516 | 0.246 | 0.186 |
|      | ImageReward | 0.774 | 0.675 | 0.785 | 0.747 | 0.699 | 0.538 | 0.423 |
|      | PickScore | 0.785 | 0.696 | 0.848 | 0.811 | 0.741 | 0.521 | 0.407 |
|      | **TextNorm** | **0.844** | **0.787** | **0.944** | **0.906** | **0.828** | **0.622** | **0.495** |

Note that while the ensemble methods above may visually resemble those proposed in the concurrent work of Coste et al. (2023), we consider an ensemble of reward models not necessarily trained on identical data or with identical hyperparameters. Moreover, we combine rewards normalized according to our proposed TextNorm method and demonstrate the approach in the context of text-to-image generation, while Coste et al. (2023) focuses on language modeling. Evaluating the effectiveness of various ensemble techniques across diverse domains would be an interesting future research.

### 4.3 QUANTITATIVE EVALUATION

We evaluate TextNorm using the metrics outlined in Section 3.2, averaged across prompts within each set in the benchmark, with the following four baselines: CLIP, BLIP-2, ImageReward, and PickScore. We use an ensemble of ImageReward and PickScore with appropriately constructed prompt sets which we release alongside the benchmark. To demonstrate both ensemble methods proposed, we use the mean ensemble for the composition set prompts and the uncertainty-regularized ensemble for prompts from the counting and comprehensive sets.

Table 3: Ablation study results.

| $X$ | $\lambda$ | AUROC | AUPRC |
|-----|-----------|-------|-------|
| ✗ | ✗ | 0.764 | 0.669 |
| Rand | ✗ | 0.641 | 0.540 |
| LLM | ✗ | 0.780 | 0.704 |
| LLM | ✓ | **0.822** | **0.752** |

Table 2 summarizes the quantitative results. We note that ImageReward and PickScore, which have been trained on considerable human feedback data, generally outperform other baseline models. Nevertheless, they exhibit weaker alignment, particularly on certain prompt types, such as those in the counting set. In comparison, TextNorm notably improves alignment across all three sets of prompts on the benchmark, as evidenced by improvements in all measured metrics.

We also conduct ablation studies to assess TextNorm based on the prompt set type used and whether an ensemble method is utilized. Table 3 summarizes the results in terms of AUROC and AUPRC averaged over all prompts in the benchmark, with PickScore as the baseline. The column $X$ denotes the use of TextNorm and specifies whether it is applied, and if so, whether a random prompt set or a set constructed with an LLM is used. The column $\lambda$ denotes whether an ensemble method is additionally used. The results indicate that (a) leveraging TextNorm with a suitable prompt set can enhance alignment, and (b) using an ensemble method can yield further improvement.

## 5 EXPERIMENTS

In this section, we assess the effectiveness of TextNorm in mitigating overoptimization across three optimization methods: best-of-$n$ sampling, supervised fine-tuning (SFT), and policy gradient-based reinforcement learning (RL). We first present qualitative comparisons in Sections 5.2 and 5.3, followed by quantitative results from human evaluations in Section 5.4.

## 5.1 SETUP

In the experiments, we use Stable Diffusion v2.1 (SD v2.1) as our base text-to-image model. For best-of-$n$ sampling, we generate a set of images using this base model and then select a subset based on the reward models for comparison. For SFT and RL fine-tuning, we use low-rank adaptation (LoRA; Hu et al. 2022) to fine-tune SD v2.1 with the reward models and compare the images generated using the fine-tuned text-to-image models. We choose the earliest checkpoint at which the number of generated images with higher scores, as measured by the reward model, than those generated using SD v2.1 reaches the maximum. Hence, a better aligned reward model improves fine-tuning by providing (a) more aligned signals for optimization and (b) a better measure based on which to select checkpoints. We apply the same parameters used for the quantitative evaluation in Section 4.3 to TextNorm for these experiments. We select and use 30 prompts from each of the comprehensive, counting, and composition sets in our benchmark for best-of-$n$ sampling, and 10 prompts from the 30 for the SFT and RL fine-tuning experiments. Further experimental details can be found in Appendix A.

## 5.2 BEST-OF-$N$ SAMPLING

We begin by evaluating all reward models using best-of-$n$ sampling, which is a simple yet effective inference-time algorithm that selects the optimal sample from a set of $n$ candidates based on a given reward model. This allows us to assess the alignment of the rewards in isolation without involving fine-tuning. Specifically, for a given prompt, we generate a set of $n \in \{16, 64, 256\}$ images using the text-to-image model and select the image with the highest reward.

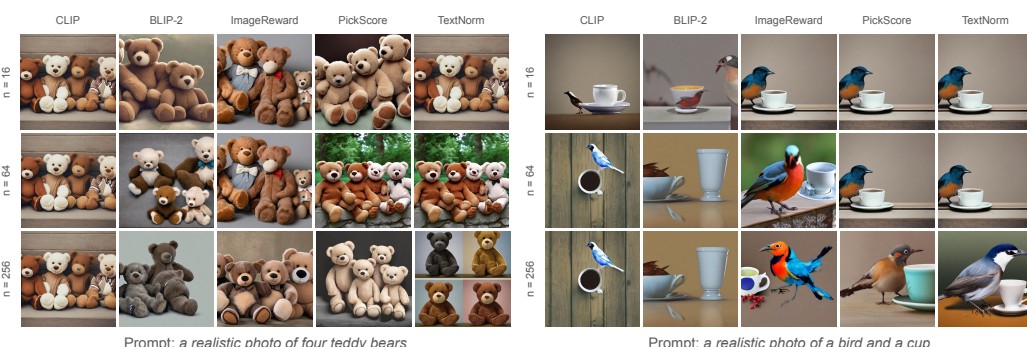

Figure 4: Images sampled using best-of-$n$ for $n \in \{16, 64, 256\}$ with the five reward models.

Figure 4 illustrates best-of-$n$ samples selected based on the five reward models. It highlights that selecting the best image from a larger pool of samples may be less desirable, especially when a poorly aligned reward model is used. Specifically, the quality of images chosen using BLIP-2 degrades as the value of $n$ increases. Other reward models generally yield better results; however, with baseline models, some selected images often depict objects that are only partially visible (see ImageReward on the left), or are arguably less realistic for higher values of $n$ (see CLIP on the right).

## 5.3 FINE-TUNING WITH REWARD MODELS

We further our evaluation by fine-tuning SD v2.1 using the reward models and comparing the images generated using the resulting models. We consider both SFT and RL-based fine-tuning.

**Supervised fine-tuning.** We adopt a setup similar to that in Lee et al. (2023), using reward-weighted regression (RWR) with the reward model scores as weights for fine-tuning. For each text prompt, we generate 100 images, then select the top 10 based on the reward model to create $\mathcal{D}^{\text{model}}$, and fine-tune SD v2.1 to maximize the reward-weighted likelihood of the data:

$$\mathcal{J}(\theta) = \mathbb{E}_{(x,y)\sim\mathcal{D}^{\text{model}}}[r(x, y)\log p_\theta(y \mid x) + \beta\log(p_\theta(y \mid x)/p_{\theta_0}(y \mid x))], \quad (4)$$

where $\beta$ is the coefficient for the Kullback-Leibler (KL) regularizer used to prevent excessive deviation from the original model $p_{\theta_0}$ (Fan et al., 2023). We experiment with multiple KL coefficients and select the one that achieves the most improvement.

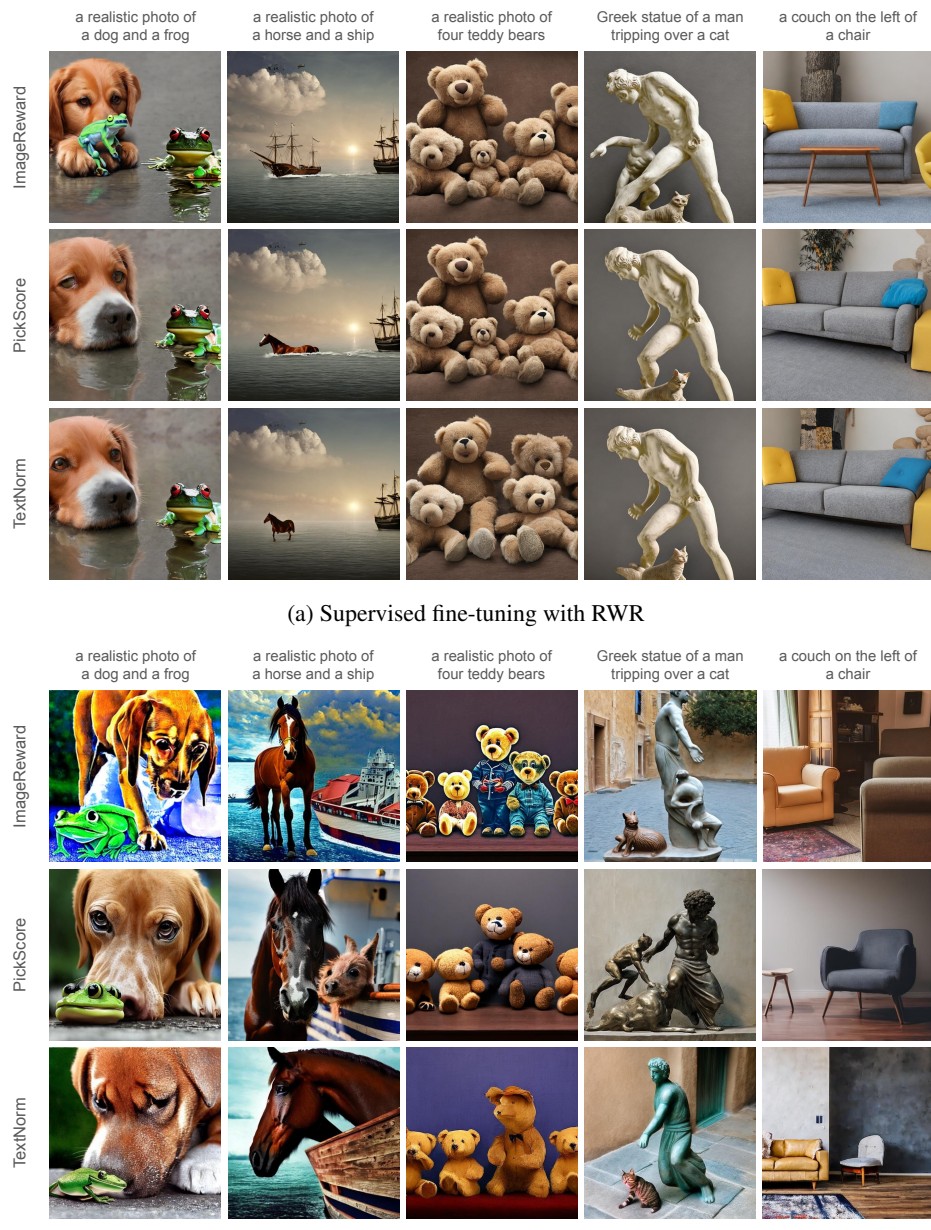

(a) Supervised fine-tuning with RWR

(b) RL fine-tuning with DDPO

Figure 5: Images generated using SD v2.1 fine-tuned with the reward models.

**RL fine-tuning.** We also consider RL-based fine-tuning, where we iteratively collect data using the current policy and perform optimization, instead of fitting a fixed distribution. We use the denoising diffusion policy optimization (DDPO; Black et al. 2023), where the denoising process is treated as a multi-step Markov decision process, allowing fine-tuning of models using policy gradient methods. Given a reward model $r$ and a prompt distribution $p(x)$, the following objective is maximized:

$$\mathcal{J}(\theta) = \mathbb{E}_{x \sim p(x), y \sim p_\theta(y|x)}[r(x, y) + \lambda(\theta_0, \theta, x, y)], \tag{5}$$

where $\lambda$ is a divergence-based regularizer to penalize deviating too far from the original model $\theta_0$.

Figure 5 shows sample images generated using the fine-tuned models. Common text-image alignment issues we observe with the models include: (a) missing an object (upper row 1, column 2), (b) containing an unmentioned object (lower row 2, column 2), (c) depicting incorrect quantities of objects (upper rows 1 and 2, column 3), and (d) disregarding the relations between objects as

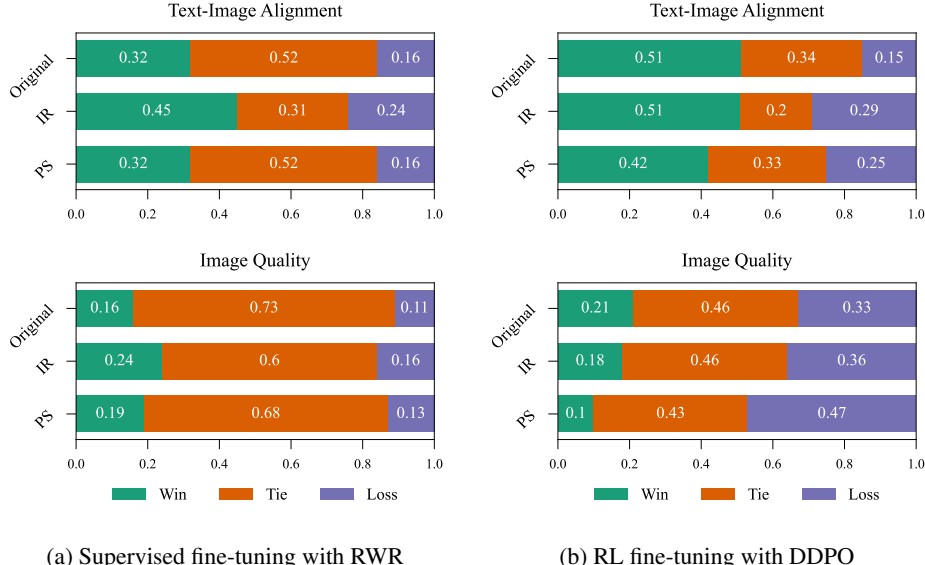

(a) Supervised fine-tuning with RWR  (b) RL fine-tuning with DDPO

Figure 6: Human evaluation results. We compare the models fine-tuned with TextNorm against SD v2.1 (Original) and those fine-tuned with ImageReward (IR) and PickScore (PS). TextNorm consistently achieves significantly better alignment with comparable image quality in the case of SFT, though with a slightly greater sacrifice in the case of RL.

described in the prompt (lower row 1, column 5). TextNorm, which aligns more closely with human judgment, provides improved rewards for optimization and a more reliable metric for selecting checkpoints, leading to images with better text-image alignment.

## 5.4 HUMAN EVALUATION

For human evaluation, we generate eight images per prompt using the fine-tuned models and create two sets of four images, resulting in a total of 60 sets evaluated by three independent annotators. Figure 6 reports the overall win, tie, and loss rates of the models fine-tuned with TextNorm compared to SD v2.1 and those fine-tuned with ImageReward and PickScore. For SFT, TextNorm achieves approximately twice as many wins for text-image alignment, with comparable or even slightly better image quality in all three comparisons. For RL, TextNorm achieves an even more dramatic improvement in text-image alignment but at the expense of image quality. We suspect that this is because in constructing the benchmark data, which are used to tune the parameters of TextNorm, the annotators primarily considered alignment in their assessment. As RL is a more powerful optimization algorithm, it possibly optimized more for alignment at the expense of image quality. Considering multiple objectives, such as alignment and quality, and leveraging the corresponding reward models in fine-tuning would be a promising approach to achieving a better balance among the objectives.

The complete human evaluation results for the best-of-$n$ sampling experiment, which also demonstrate the effectiveness of TextNorm, are provided in Appendix C.

## 6 CONCLUSION

Fine-tuning models on a reward function trained on human feedback data has emerged as a promising method for aligning models with human intent. However, excessive optimization can degrade model performance. In this work, we introduce the TIA2 benchmark to assess several state-of-the-art reward models in text-to-image generation. We find that the reward models, even those trained on human data, are often not well-aligned with human assessment. We demonstrate that overoptimization occurs particularly when poorly aligned reward models are used for fine-tuning. To address this issue, we introduce TextNorm, a simple method that enhances alignment through reward calibration using semantically contrastive prompts. We demonstrate both quantitatively and qualitatively that the confidence-calibrated scores effectively reduces overoptimization.

ETHICS STATEMENT

Generative models for image synthesis, like many machine learning models in general, are susceptible to learning biases inherent in the training data (Birhane et al., 2021). While fine-tuning pre-trained models with a suitable reward can enhance the models, overoptimzing against an imperfect proxy objective can instead degrade model quality, as we investigate in depth in this work. Hence, it is important to understand the limitations of both the pre-trained models and the rewards with which they are fine-tuned. Our work is a timely exploration of this issue in text-to-image generation, examining various state-of-the-art reward models and reporting their limitations as both evaluation metrics and training objectives. We also introduce a simple method for better aligning reward models with human intent. While we demonstrate the effectiveness of our method in reducing overoptimization, the implementation relies on external models such as an LLM and a VLM. Therefore, careful selection of these models is crucial, as the success of the implementation hinges on their quality.

REPRODUCIBILITY STATEMENT

We provide implementation details including the experiment setup, the models used, and the training specifications (e.g., hyperparameters) in Section 5 and in Appendix A.

ACKNOWLEDGMENTS

We thank Seunghwan Lee, Sojeong Kim, Dongjun Lee, Changyeon Kim, Jongjin Park, Yisol Choi, Hyunsub Jeong, Junesuk Choi, and Younghyun Kim for their help with labeling the data of the main experiments of this work.

This work was supported by Institute of Information & communications Technology Planning & Evaluation (IITP) grant funded by the Korea government (MSIT) (No.2019-0-00075, Artificial Intelligence Graduate School Program (KAIST); No.2021-0-02068, Artificial Intelligence Innovation Hub; No.2022-0-00959, Few-shot Learning of Casual Inference in Vision and Language for Decision Making).

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

# A    EXPERIMENTAL DETAILS

## A.1    BASELINE REWARD MODELS

We use publicly available implementations of the baseline reward models. For CLIP, we use the CLIP ViT-B/32 model from the official CLIP implementation provided by OpenAI.[2] For BLIP-2, we use the `pretrain` version of the `blip2` model from the Salesforce LAVIS library.[3] For ImageReward[4] and PickScore,[5] we use the official models provided by the authors of the papers, which are based on the BLIP and CLIP ViT-H/14 architectures, respectively.

## A.2    FINE-TUNING TEXT-TO-IMAGE MODELS

Our fine-tuning implementations are based on publicly available Python libraries. Specifically, we used the diffusers library[6] for the SFT experiments and the trl library[7] for the RL experiments. For SFT, we used `stabilityai/stable-diffusion-2-1` as our base model and fine-tuned it for up to 1,000 steps. We experimented with KL coefficients ranging from 0.00 to 1.00, in increments of 0.25, and selected the one that yielded the most improvement. For RL, we used `stabilityai/stable-diffusion-2-1-base` as our base model and fine-tuned it for up to 500 epochs over the training prompts. Note that we have conducted minimal hyperparameter tuning and, in many cases, adopted the default values provided by the library code.

Table 4: Summary of hyperparameters used for SFT and RL fine-tuning.

|  | Parameters | RWR | DDPO |
|---|---|---|---|
| Diffusion | Denoising steps | 50 | 50 |
|  | Guidance scale | 7.5 | 5.0 |
| Optimization | Optimizer | AdamW | AdamW |
|  | Learning rate | 1e-5 | 3e-4 |
|  | Weight decay | 1e-2 | 1e-4 |
|  | $\beta_1$ | 0.9 | 0.9 |
|  | $\beta_2$ | 0.999 | 0.999 |
|  | $\epsilon$ | 1e-8 | 1e-8 |
|  | Max gradient norm | 1.0 | 1.0 |
| Training | Batch size | 32 | 64 |
|  | Samples per iteration | - | 256 |
|  | Gradient updates per iteration | - | 1 |
|  | Mixed precision | fp16 | fp16 |

## A.3    PARAMETERS FOR TEXTNORM

The TextNorm score used for both the quantitative analysis in Section 4.3 and the experiments in Section 5 is derived from an ensemble of ImageReward and PickScore. To demonstrate both ensemble methods discussed in Section 4.2, we employed the mean ensemble for the composition set prompts and the uncertainty-regularized ensemble for prompts from the counting and comprehensive sets. We only lightly tuned the temperature parameter and the uncertainty penalty coefficient based on the average of the three AP metrics considered in this work.

---

[2]https://github.com/openai/CLIP
[3]https://github.com/salesforce/LAVIS
[4]https://github.com/THUDM/ImageReward
[5]https://github.com/yuvalkirstain/PickScore
[6]https://github.com/huggingface/diffusers
[7]https://github.com/huggingface/trl

# B DETAILS ON THE TIA2 BENCHMARK

## B.1 EXAMPLES AND STATISTICS

Table 5 provides sample prompts from each of the three sets in the benchmark.

Table 5: Sample prompts taken from TIA2 per category.

| Category | Examples |
|---|---|
| Comprehensive | `minnie mouse and baby shark cartoon`
`A 1960s poster warning against climate change`
`cute bee wearing chef hat colored pencil art` |
| Counting | `a realistic photo of three dogs`
`a realistic photo of four deer`
`a realistic photo of six teddy bears` |
| Composition | `a realistic photo of a deer and a truck`
`a realistic photo of a bird and an umbrella`
`a realistic photo of an airplane and a ship` |

Table 6 reports the complete set of object classes used to create the synthetic prompts. Table 7 provides further statistics on the subcategories of the comprehensive set.

Table 6: Object class names used to generate synthetic prompts in TIA2.

| Subcategory | Object classes |
|---|---|
| Accessory | suitcase, umbrella |
| Appliance | microwave, toaster |
| Animal | bird, cat, dog, horse, deer, frog |
| Electronic | laptop |
| Food | cake, apple, orange, carrot |
| Furniture | chair |
| Indoor | book, teddy bear, vase |
| Kitchen | cup, fork |
| Vehicle | automobile, airplane, truck, ship |

Table 7: Statistics on each subcategory of the comprehensive set in TIA2.

| Subcategory | Counts | | Human labels (%) | |
|---|---|---|---|---|
| | Texts | Images | Good | Bad |
| Colors | 10 | 500 | 91.20 | 8.80 |
| Composition | 10 | 500 | 33.00 | 67.00 |
| Counting | 10 | 500 | 52.80 | 47.20 |
| Creative | 10 | 500 | 70.00 | 30.00 |
| Location | 10 | 500 | 84.20 | 15.80 |
| Reddit | 20 | 1,000 | 36.60 | 63.40 |
| Spatial | 10 | 500 | 14.20 | 85.80 |
| Style | 10 | 500 | 53.20 | 46.80 |
| Text | 10 | 500 | 0.40 | 99.60 |
| Total | 100 | 5,000 | 47.22 | 52.78 |

## B.2 LABELING PROCEDURE

For each text-image pair in the benchmark, we asked three human annotators to assign a binary label indicating whether the text and image are semantically aligned, or to indicate that the assessment is inconclusive. We asked the annotators to assess text-image alignment first and foremost, and to consider image fidelity at their discretion if it was noticeable enough to affect alignment. Table 8 shows an excerpt of the labeling instructions we provided to the annotators for the task. Figure 7 is a screenshot of the labeling interface the annotators used.

Table 8: Excerpt from the labeling instructions given to annotators.

| **Instruction** |
|---|
| Options = {1: Good, 2: Bad and 3: Skip (hard to answer)} |
| (1) If the provided prompt aligns well with the image, select 1; otherwise, select 2. |
| (2) If the quality of the image is so poor that it affects the alignment, select 2. |
| (3) If you feel you mislabeled something, you can relabel it. |

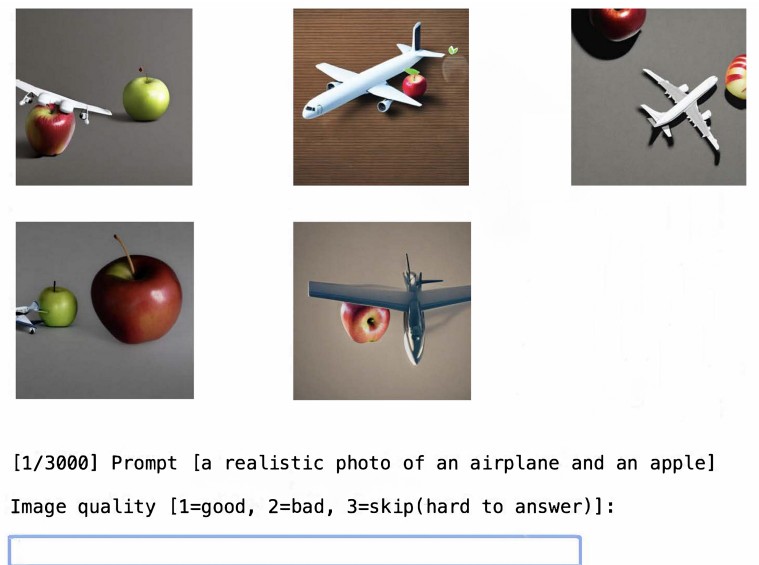

[1/3000] Prompt [a realistic photo of an airplane and an apple]

Image quality [1=good, 2=bad, 3=skip(hard to answer)]:

Figure 7: Screenshot of the labeling interface.

### B.3 ADDITIONAL ANALYSIS

We conduct additional analysis on the reward models, reporting the average AUROC for each synthetic prompt included in the benchmark. Since each synthetic prompt corresponds to either a pair of two object classes or a combination of an object class and a count, this analysis offers deeper insights into how well the reward models align with human judgment, depending on the type and quantity of object classes considered in the dataset.

In the tables below, we present heat maps displaying the per-prompt average AUROC values. Cells with a value of 0.75 remain uncolored, while those with higher values are shaded in varying intensities of blue, indicating higher scores. Conversely, cells with lower values are shaded in varying intensities of red, denoting lower scores. As the heat maps suggest, both ImageReward and PickScore generally attain higher AUROC values compared to CLIP and BLIP-2, with TextNorm consistently outperforming all baselines.

Table 9: Per-prompt AUROC for CLIP on the composition set.

| | airplane | apple | automobile | bird | book | cake | carrot | cat | chair | cup | deer | dog | fork | frog | horse | laptop | microwave | orange | ship | suitcase | teddy bear | toaster | truck | umbrella | vase |
|---|---|---|---|---|---|---|---|---|---|---|---|---|---|---|---|---|---|---|---|---|---|---|---|---|---|
| airplane | | 0.73 | 0.79 | 0.57 | 0.6 | 0.58 | 0.93 | 0.76 | 0.94 | 0.66 | 0.84 | 0.89 | 1 | 0.88 | 0.77 | 0.64 | 0.78 | 0.69 | 0.59 | 0.57 | 0.7 | 0.58 | 0.69 | 0.82 | 0.87 |
| apple | 0.73 | | 0.73 | 0.69 | 0.62 | 0.74 | 0.92 | 0.69 | 0.75 | 0.87 | 0.84 | 0.68 | 0.81 | 0.66 | 0.78 | 0.68 | 0.77 | 0.78 | 0.72 | 0.78 | 0.6 | 0.73 | 0.77 | 0.75 | 0.91 |
| automobile | 0.79 | 0.73 | | 0.63 | 0.86 | 0.6 | 0.68 | 0.73 | 0.89 | 0.85 | 0.79 | 0.69 | 0.82 | 0.59 | 0.58 | 0.87 | 0.88 | 0.79 | 0.62 | 0.71 | 0.73 | 0.47 | 0.61 | 0.68 | 0.78 |
| bird | 0.57 | 0.69 | 0.63 | | 0.72 | 0.7 | 0.73 | 0.94 | 0.69 | 0.65 | | 0.91 | 0.73 | 0.7 | 0.9 | 0.55 | 0.55 | 0.66 | 0.66 | 0.54 | 0.93 | 0.76 | 0.7 | 0.72 | 0.78 |
| book | 0.6 | 0.62 | 0.86 | 0.72 | | 0.68 | 0.67 | 0.62 | 0.72 | 0.49 | 0.46 | 0.67 | 0.83 | 0.51 | 0.58 | 0.68 | 0.65 | 0.79 | 0.91 | 0.83 | 0.59 | 0.68 | 0.85 | 0.63 | 0.7 |
| cake | 0.58 | 0.74 | 0.6 | 0.7 | 0.68 | | 0.64 | 0.78 | 0.84 | 0.78 | 0.57 | 0.71 | 0.57 | 0.66 | 0.73 | 0.57 | 0.71 | 0.57 | 0.64 | 0.78 | 0.75 | 0.87 | 0.65 | 0.8 | 0.77 |
| carrot | 0.93 | 0.92 | 0.68 | 0.73 | 0.67 | 0.64 | | 0.57 | 0.72 | 0.64 | 0.79 | 0.79 | 0.69 | 0.75 | 0.79 | 0.61 | 0.68 | 0.5 | 0.93 | 0.67 | 0.67 | 0.63 | 0.71 | 0.7 | 0.76 |
| cat | 0.76 | 0.69 | 0.73 | 0.94 | 0.62 | 0.78 | 0.57 | | 0.56 | 0.55 | | 0.88 | 0.74 | 0.75 | 0.88 | 0.65 | 0.65 | 0.56 | 0.65 | 0.55 | 0.77 | 0.64 | 0.67 | 0.95 | 0.65 |
| chair | 0.94 | 0.75 | 0.89 | 0.69 | 0.72 | 0.84 | 0.72 | 0.56 | | 0.97 | 0.81 | 0.7 | 0.97 | 0.78 | 0.79 | 0.78 | 0.77 | 0.9 | 0.83 | 0.87 | 0.76 | 0.7 | 0.89 | 0.7 | 0.77 |
| cup | 0.66 | 0.87 | 0.85 | 0.65 | 0.49 | 0.78 | 0.64 | 0.55 | 0.97 | | 0.73 | 0.52 | 0.79 | 0.65 | 0.58 | 0.82 | 0.78 | 0.74 | 0.67 | 0.77 | 0.49 | 0.7 | 0.94 | 0.72 | 0.65 |
| deer | 0.84 | 0.84 | 0.79 | | 0.46 | 0.57 | 0.79 | | 0.81 | 0.73 | | 0.8 | 0.9 | 0.88 | 0.64 | 0.77 | 0.6 | 0.59 | 0.73 | 0.53 | 0.82 | 0.69 | 0.68 | 0.7 | 0.76 |
| dog | 0.89 | 0.68 | 0.69 | 0.91 | 0.66 | 0.71 | 0.79 | 0.88 | 0.7 | 0.52 | 0.8 | | 0.46 | 0.62 | 0.8 | 0.57 | 0.56 | 0.63 | 0.74 | 0.51 | 0.92 | 0.65 | 0.66 | 0.46 | 0.61 |
| fork | 1 | 0.81 | 0.82 | 0.73 | 0.83 | 0.57 | 0.69 | 0.74 | 0.97 | 0.79 | 0.9 | 0.46 | | 0.59 | 0.91 | 0.86 | 0.86 | 0.52 | | 0.77 | 0.86 | 0.79 | 0.94 | 0.93 | 0.83 |
| frog | 0.88 | 0.66 | 0.59 | 0.7 | 0.51 | 0.66 | 0.75 | 0.75 | 0.78 | 0.65 | 0.88 | 0.62 | 0.59 | | 0.91 | 0.53 | 0.57 | 0.68 | 0.8 | 0.54 | 0.84 | 0.73 | 0.74 | 0.82 | 0.69 |
| horse | 0.77 | 0.78 | 0.58 | 0.9 | 0.68 | 0.58 | 0.73 | 0.79 | 0.88 | 0.79 | 0.58 | 0.64 | 0.8 | 0.91 | | 0.91 | 0.45 | 0.65 | 0.7 | 0.85 | 0.66 | 0.66 | 0.63 | 0.61 | 0.85 |
| laptop | 0.64 | 0.68 | 0.87 | 0.55 | 0.68 | 0.57 | 0.61 | 0.65 | 0.78 | 0.82 | 0.77 | 0.57 | 0.86 | 0.53 | 0.45 | | 0.77 | 0.46 | 0.83 | 0.66 | 0.68 | | 0.81 | 0.55 | 0.7 |
| microwave | 0.78 | 0.77 | 0.88 | 0.55 | 0.65 | 0.71 | 0.68 | 0.65 | 0.77 | 0.78 | 0.6 | 0.56 | 0.68 | 0.57 | 0.65 | 0.77 | | 0.79 | | 0.74 | 0.84 | 0.87 | 0.63 | 0.82 | 0.73 |
| orange | 0.69 | 0.78 | 0.79 | 0.66 | 0.79 | 0.57 | 0.5 | 0.56 | 0.9 | 0.74 | 0.59 | 0.63 | 0.52 | 0.68 | 0.7 | 0.46 | 0.79 | | 0.74 | 0.64 | 0.68 | 0.74 | 0.69 | 0.93 | 0.69 |
| ship | 0.59 | 0.72 | 0.62 | 0.66 | 0.91 | 0.64 | 0.93 | 0.65 | 0.83 | 0.67 | 0.73 | 0.74 | | 0.8 | 0.85 | 0.83 | 0.84 | 0.74 | | 0.71 | 0.76 | 0.73 | 0.85 | 0.64 | 0.59 |
| suitcase | 0.57 | 0.78 | 0.71 | 0.54 | 0.83 | 0.78 | 0.67 | 0.55 | 0.87 | 0.77 | 0.53 | 0.51 | 0.77 | 0.54 | 0.66 | 0.66 | 0.87 | 0.64 | 0.71 | | 0.52 | 0.62 | 0.79 | 0.67 | 0.48 |
| teddy bear | 0.7 | 0.6 | 0.73 | 0.93 | 0.59 | 0.75 | 0.67 | 0.77 | 0.76 | 0.49 | 0.82 | 0.92 | 0.86 | 0.84 | 0.66 | 0.68 | 0.63 | 0.68 | 0.76 | 0.52 | | 0.61 | 0.6 | 0.72 | 0.61 |
| toaster | 0.58 | 0.73 | 0.47 | 0.76 | 0.68 | 0.87 | 0.63 | 0.64 | 0.7 | 0.7 | 0.69 | 0.65 | 0.79 | 0.73 | 0.63 | | 0.74 | 0.73 | 0.62 | 0.61 | | | 0.92 | 0.86 | 0.47 |
| truck | 0.69 | 0.77 | 0.61 | 0.7 | 0.85 | 0.65 | 0.71 | 0.67 | 0.89 | 0.94 | 0.68 | 0.66 | 0.94 | 0.74 | 0.61 | 0.81 | 0.82 | 0.69 | 0.85 | 0.79 | 0.6 | 0.92 | | 0.64 | 0.91 |
| umbrella | 0.82 | 0.75 | 0.68 | 0.72 | 0.63 | 0.8 | 0.7 | 0.95 | 0.7 | 0.72 | 0.7 | 0.46 | 0.93 | 0.82 | 0.61 | 0.55 | 0.79 | 0.93 | 0.64 | 0.67 | 0.72 | 0.86 | 0.64 | | 0.44 |
| vase | 0.87 | 0.91 | 0.78 | 0.78 | 0.7 | 0.77 | 0.76 | 0.65 | 0.77 | 0.65 | 0.76 | 0.61 | 0.83 | 0.69 | 0.85 | 0.7 | 0.73 | 0.69 | 0.59 | 0.48 | 0.61 | 0.47 | 0.91 | 0.44 | |

Table 10: Per-prompt AUROC for BLIP-2 on the composition set.

| | airplane | apple | automobile | bird | book | cake | carrot | cat | chair | cup | deer | dog | fork | frog | horse | laptop | microwave | orange | ship | suitcase | teddy bear | toaster | truck | umbrella | vase |
|---|---|---|---|---|---|---|---|---|---|---|---|---|---|---|---|---|---|---|---|---|---|---|---|---|---|
| airplane | | 0.52 | 0.78 | 0.7 | 0.47 | 0.61 | 0.59 | 0.63 | 0.64 | 0.37 | 0.67 | 0.71 | 0.75 | 0.76 | 0.76 | 0.48 | 0.71 | 0.6 | 0.55 | 0.58 | 0.43 | 0.88 | 0.7 | 0.65 | 0.77 |
| apple | 0.52 | | 0.64 | 0.5 | 0.57 | 0.71 | 0.92 | 0.57 | 0.62 | 0.81 | 0.67 | 0.55 | 0.68 | 0.62 | 0.78 | 0.51 | 0.48 | 0.55 | 0.65 | 0.79 | 0.72 | 0.68 | 0.7 | 0.6 | 0.9 |
| automobile | 0.78 | 0.64 | | 0.51 | 0.79 | 0.52 | 0.53 | 0.53 | 0.84 | 0.74 | 0.64 | 0.47 | 0.7 | 0.61 | 0.48 | 0.78 | 0.72 | 0.62 | 0.61 | 0.72 | 0.52 | 0.68 | 0.7 | 0.58 | 0.63 |
| bird | 0.7 | 0.5 | 0.51 | | 0.45 | 0.42 | 0.58 | 0.94 | 0.55 | 0.48 | | 0.65 | 0.74 | 0.58 | 0.62 | 0.47 | 0.43 | 0.54 | 0.48 | 0.39 | 0.94 | 0.45 | 0.51 | 0.48 | 0.59 |
| book | 0.47 | 0.57 | 0.79 | 0.45 | | 0.64 | 0.56 | 0.56 | 0.77 | 0.27 | 0.73 | 0.34 | 0.64 | 0.59 | 0.54 | 0.55 | 0.52 | 0.57 | 0.56 | 0.62 | 0.45 | 0.57 | 0.78 | 0.48 | 0.57 |
| cake | 0.61 | 0.71 | 0.52 | 0.42 | 0.64 | | 0.55 | 0.4 | 0.78 | 0.49 | 0.49 | 0.36 | 0.5 | 0.42 | 0.66 | 0.4 | 0.72 | 0.65 | 0.63 | 0.54 | 0.62 | 0.62 | 0.63 | 0.63 | 0.67 |
| carrot | 0.59 | 0.92 | 0.53 | 0.58 | 0.56 | 0.55 | | 0.63 | 0.64 | 0.63 | 0.63 | 0.69 | 0.57 | 0.54 | 0.79 | 0.68 | 0.66 | 0.72 | 0.91 | 0.6 | 0.59 | 0.58 | 0.78 | 0.6 | 0.67 |
| cat | 0.63 | 0.57 | 0.53 | 0.94 | 0.56 | 0.4 | 0.63 | | 0.44 | 0.31 | | 0.57 | 0.55 | 0.53 | 0.74 | 0.55 | 0.43 | 0.47 | 0.55 | 0.45 | 0.49 | 0.35 | 0.5 | 0.71 | 0.28 |
| chair | 0.64 | 0.62 | 0.84 | 0.55 | 0.77 | 0.78 | 0.64 | 0.44 | | 0.85 | 0.7 | 0.45 | 0.54 | 0.57 | 0.57 | 0.82 | 0.82 | 0.61 | 0.65 | 0.83 | 0.61 | 0.78 | 0.9 | 0.45 | 0.64 |
| cup | 0.37 | 0.81 | 0.74 | 0.48 | 0.27 | 0.49 | 0.63 | 0.31 | 0.85 | | 0.51 | 0.32 | 0.82 | 0.25 | 0.47 | 0.67 | 0.46 | 0.64 | 0.66 | 0.72 | 0.49 | 0.66 | 0.8 | 0.62 | 0.65 |
| deer | 0.67 | 0.67 | 0.64 | | 0.73 | 0.49 | 0.63 | | 0.7 | 0.51 | | 0.75 | 0.81 | 0.72 | 0.55 | 0.45 | 0.49 | 0.52 | 0.5 | 0.57 | 0.71 | 0.51 | 0.53 | 0.47 | 0.68 |
| dog | 0.71 | 0.55 | 0.47 | 0.65 | 0.34 | 0.36 | 0.69 | 0.57 | 0.45 | 0.32 | 0.75 | | 0.6 | 0.54 | 0.72 | 0.41 | 0.29 | 0.44 | 0.64 | 0.35 | 0.84 | 0.66 | 0.62 | 0.44 | 0.45 |
| fork | 0.75 | 0.68 | 0.7 | 0.74 | 0.64 | 0.5 | 0.57 | 0.55 | 0.54 | 0.82 | 0.81 | 0.6 | | 0.39 | 0.89 | 0.8 | 0.86 | 0.62 | | 0.76 | 0.77 | 0.9 | 0.77 | 0.89 | 0.87 |
| frog | 0.76 | 0.62 | 0.61 | 0.58 | 0.59 | 0.42 | 0.54 | 0.53 | 0.57 | 0.25 | 0.72 | 0.54 | 0.39 | | 0.69 | 0.51 | 0.47 | 0.41 | 0.68 | 0.41 | 0.6 | 0.48 | 0.59 | 0.63 | 0.57 |
| horse | 0.76 | 0.78 | 0.48 | 0.62 | 0.54 | 0.66 | 0.79 | 0.74 | 0.57 | 0.47 | 0.55 | 0.72 | 0.89 | 0.69 | | 0.33 | 0.61 | 0.56 | 0.81 | 0.54 | 0.57 | 0.47 | 0.58 | 0.34 | 0.76 |
| laptop | 0.48 | 0.51 | 0.78 | 0.47 | 0.55 | 0.4 | 0.68 | 0.55 | 0.82 | 0.67 | 0.41 | 0.8 | 0.51 | 0.33 | | | 0.84 | 0.55 | 0.69 | 0.55 | 0.62 | | 0.62 | 0.44 | 0.61 |
| microwave | 0.71 | 0.48 | 0.72 | 0.43 | 0.52 | 0.72 | 0.66 | 0.43 | 0.82 | 0.46 | 0.49 | 0.29 | 0.86 | 0.47 | 0.61 | 0.84 | | 0.55 | 0.78 | 0.81 | 0.5 | | 1 | 0.73 | 0.5 |
| orange | 0.6 | 0.55 | 0.62 | 0.54 | 0.57 | 0.65 | 0.72 | 0.47 | 0.61 | 0.64 | 0.52 | 0.44 | 0.62 | 0.41 | 0.56 | 0.55 | 0.55 | | 0.74 | 0.62 | 0.68 | 0.71 | 0.62 | 0.8 | 0.54 |
| ship | 0.55 | 0.65 | 0.61 | 0.48 | 0.56 | 0.63 | 0.91 | 0.55 | 0.65 | 0.66 | 0.5 | 0.64 | | 0.68 | 0.81 | 0.69 | 0.78 | 0.74 | | 0.76 | 0.58 | 0.62 | 0.8 | 0.68 | 0.94 |
| suitcase | 0.58 | 0.79 | 0.72 | 0.39 | 0.64 | 0.54 | 0.6 | 0.45 | 0.83 | 0.72 | 0.57 | 0.35 | 0.76 | 0.41 | 0.54 | 0.55 | 0.81 | 0.62 | 0.76 | | 0.58 | 0.62 | 0.8 | 0.5 | 0.54 |
| teddy bear | 0.43 | 0.72 | 0.52 | 0.94 | 0.45 | 0.62 | 0.59 | 0.49 | 0.61 | 0.49 | 0.71 | 0.84 | 0.77 | 0.6 | 0.57 | 0.62 | 0.5 | 0.68 | 0.72 | 0.58 | | 0.41 | 0.41 | 0.46 | 0.67 |
| toaster | 0.88 | 0.68 | 0.68 | 0.45 | 0.57 | 0.62 | 0.58 | 0.35 | 0.78 | 0.66 | 0.51 | 0.46 | 0.9 | 0.48 | 0.47 | | 0.71 | | 0.99 | 0.62 | 0.41 | | 0.9 | 0.75 | 0.53 |
| truck | 0.7 | 0.7 | 0.7 | 0.51 | 0.78 | 0.62 | 0.78 | 0.5 | 0.9 | 0.8 | 0.53 | 0.62 | 0.77 | 0.59 | 0.58 | 0.62 | 1 | 0.74 | 0.8 | 0.41 | 0.9 | | | 0.57 | 0.85 |
| umbrella | 0.65 | 0.6 | 0.58 | 0.48 | 0.48 | 0.63 | 0.6 | 0.71 | 0.45 | 0.62 | 0.47 | 0.44 | 0.89 | 0.63 | 0.34 | 0.44 | 0.73 | 0.8 | 0.68 | 0.5 | 0.46 | 0.75 | 0.57 | | 0.61 |
| vase | 0.77 | 0.9 | 0.63 | 0.59 | 0.57 | 0.67 | 0.67 | 0.28 | 0.64 | 0.65 | 0.68 | 0.45 | 0.87 | 0.57 | 0.76 | 0.61 | 0.5 | 0.51 | 0.94 | 0.54 | 0.67 | 0.53 | 0.85 | 0.61 | |

Table 11: Per-prompt AUROC for ImageReward on the composition set.

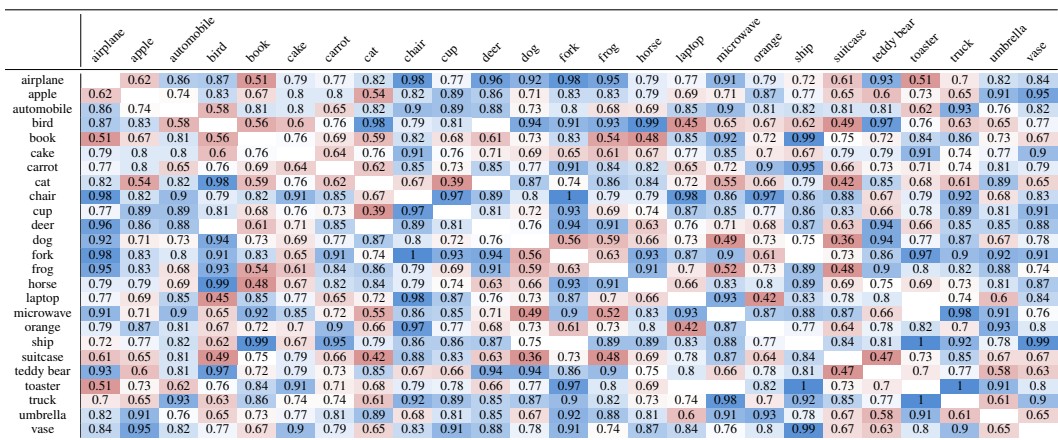

Table 12: Per-prompt AUROC for PickScore on the composition set.

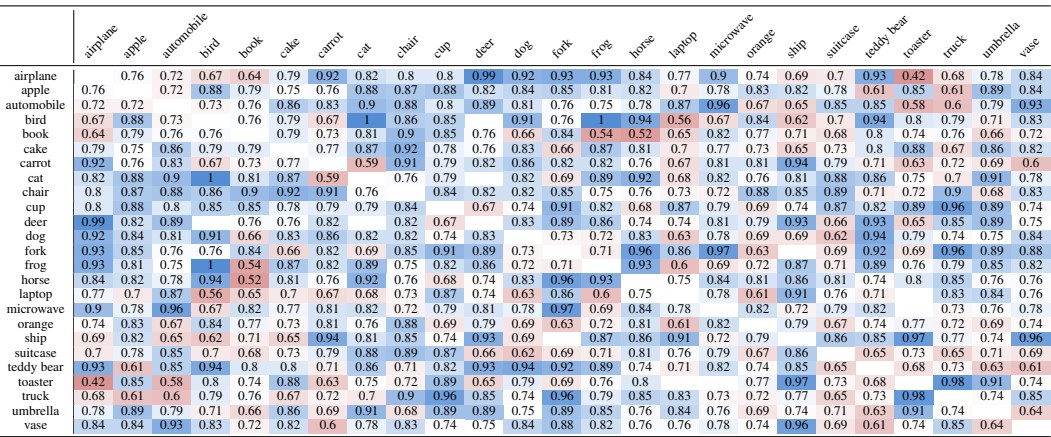

Table 13: Per-prompt AUROC for TextNorm on the composition set.

| | airplane | apple | automobile | bird | book | cake | carrot | cat | chair | cup | deer | dog | fork | frog | horse | laptop | microwave | orange | ship | suitcase | teddy bear | toaster | truck | umbrella | vase |
|---|---|---|---|---|---|---|---|---|---|---|---|---|---|---|---|---|---|---|---|---|---|---|---|---|---|
| airplane | | 0.79 | 0.85 | 0.78 | 0.51 | 0.85 | 0.89 | 0.83 | 0.99 | 0.79 | 0.97 | 0.96 | 1 | 0.96 | 0.83 | 0.96 | 0.95 | 0.79 | 0.87 | 0.86 | 0.93 | 0.63 | 0.78 | 0.86 | 0.9 |
| apple | 0.79 | | 0.8 | 0.89 | 0.82 | 0.86 | 0.9 | 0.87 | 0.9 | 0.97 | 0.88 | 0.92 | 0.89 | 0.83 | 0.89 | 0.57 | 0.73 | 0.89 | 0.87 | 0.9 | 0.7 | 0.83 | 0.81 | 0.9 | 0.99 |
| automobile | 0.85 | 0.8 | | 0.77 | 0.74 | 0.85 | 0.84 | 0.91 | 0.94 | 0.92 | 0.86 | 0.9 | 0.92 | 0.77 | 0.85 | 0.92 | 0.94 | 0.71 | 0.88 | 0.86 | 0.89 | 0.67 | 0.88 | 0.86 | 0.96 |
| bird | 0.78 | 0.89 | 0.77 | | 0.69 | 0.84 | 0.77 | 0.73 | 1 | 0.77 | 0.85 | 0.99 | 0.92 | 0.97 | 1 | 0.44 | 0.69 | 0.84 | 0.77 | 0.61 | 0.99 | 0.81 | 0.84 | 0.67 | 0.82 |
| book | 0.51 | 0.82 | 0.74 | 0.69 | | 0.84 | 0.77 | 0.73 | 0.86 | 0.81 | 0.63 | 0.75 | 0.89 | 0.8 | 0.48 | 0.84 | 0.98 | 0.9 | 0.89 | 0.85 | 0.86 | 0.86 | 0.83 | 0.7 | 0.77 |
| cake | 0.85 | 0.86 | 0.85 | 0.71 | 0.84 | | 0.66 | 0.92 | 0.92 | 0.86 | 0.83 | 0.82 | 0.77 | 0.79 | 0.79 | 0.89 | | 0.75 | 0.72 | 0.91 | 0.84 | 0.89 | 0.82 | 0.84 | 0.94 |
| carrot | 0.89 | 0.9 | 0.84 | 0.71 | 0.77 | 0.66 | | 0.72 | 0.91 | 0.92 | 0.92 | 0.88 | 0.93 | 0.89 | 0.88 | 0.82 | 0.88 | 0.98 | 0.99 | 0.8 | 0.8 | 0.9 | 0.85 | 0.83 | 0.82 |
| cat | 0.83 | 0.87 | 0.91 | 1 | 0.73 | 0.92 | 0.72 | | 0.77 | 0.57 | | 0.89 | 0.77 | 0.94 | 0.88 | 0.72 | 0.71 | 0.73 | 0.85 | 0.77 | 0.87 | 0.82 | 0.69 | 0.92 | 0.73 |
| chair | 0.99 | 0.9 | 0.94 | 0.77 | 0.86 | 0.92 | 0.91 | 0.77 | | 0.98 | 0.89 | 0.93 | 1 | 0.76 | 0.78 | 0.95 | 0.86 | 0.97 | 0.89 | 0.98 | 0.84 | 0.82 | 0.94 | 0.86 | 0.9 |
| cup | 0.79 | 0.97 | 0.92 | 0.85 | 0.81 | 0.86 | 0.92 | 0.57 | 0.98 | | 0.84 | 0.81 | 0.97 | 0.92 | 0.76 | 0.89 | 0.91 | 0.9 | 0.97 | 0.92 | 0.78 | 0.82 | 0.95 | 0.85 | 0.89 |
| deer | 0.97 | 0.88 | 0.86 | 0.63 | 0.83 | 0.92 | | 0.89 | 0.84 | | | 0.92 | 0.93 | 0.93 | 0.68 | 0.77 | 0.79 | 0.82 | 0.89 | 0.9 | 0.95 | 0.65 | 0.9 | 0.92 | 0.86 |
| dog | 0.96 | 0.92 | 0.9 | 0.99 | 0.75 | 0.82 | 0.88 | 0.89 | 0.93 | 0.81 | 0.92 | | 0.71 | 0.84 | 0.97 | 0.89 | 0.94 | 0.86 | 0.8 | 0.63 | 0.89 | 0.85 | 0.89 | 0.81 | 0.82 |
| fork | 1 | 0.89 | 0.82 | 0.92 | 0.89 | 0.77 | 0.93 | 0.77 | 1 | 0.97 | 0.93 | 0.71 | | 0.84 | 0.97 | 0.89 | 0.94 | 0.86 | | 0.75 | 0.91 | 0.88 | 0.99 | 0.99 | 0.95 |
| frog | 0.96 | 0.83 | 0.77 | 0.8 | 0.79 | 0.89 | 0.94 | 0.76 | 0.92 | 0.93 | 0.78 | 0.84 | | | 1 | 0.63 | 0.59 | 0.81 | 0.94 | 0.68 | 0.91 | 0.78 | 0.89 | 0.89 | 0.83 |
| horse | 0.83 | 0.89 | 0.85 | 1 | 0.48 | 0.79 | 0.88 | 0.88 | 0.78 | 0.76 | 0.68 | 0.75 | 0.97 | 1 | | 0.8 | 0.85 | 0.92 | 0.89 | 0.8 | 0.82 | 0.71 | 0.85 | 0.88 | 0.9 |
| laptop | 0.96 | 0.57 | 0.92 | 0.44 | 0.84 | 0.82 | 0.82 | 0.72 | 0.95 | 0.89 | 0.77 | 0.76 | 0.89 | 0.63 | 0.8 | | 0.94 | 0.81 | 0.88 | 0.82 | 0.67 | | 0.86 | 0.75 | 0.87 |
| microwave | 0.95 | 0.73 | 0.94 | 0.69 | 0.98 | 0.89 | 0.88 | 0.71 | 0.86 | 0.91 | 0.79 | 0.62 | 0.94 | 0.59 | 0.85 | 0.94 | | 0.89 | 0.94 | 0.75 | 0.85 | 0.88 | 0.86 | 0.86 | 0.88 |
| orange | 0.79 | 0.89 | 0.71 | 0.84 | 0.9 | 0.75 | 0.98 | 0.73 | 0.97 | 0.9 | 0.82 | 0.84 | 0.86 | 0.81 | 0.92 | 0.81 | 0.89 | | 0.94 | 0.75 | 0.85 | 0.88 | 0.72 | 0.95 | 0.99 |
| ship | 0.87 | 0.87 | 0.88 | 0.77 | 0.89 | 0.72 | 0.99 | 0.85 | 0.89 | 0.97 | 0.89 | 0.8 | | 0.94 | 0.89 | 0.88 | 0.94 | 0.94 | | 0.91 | 0.91 | 0.99 | 0.93 | 0.74 | 0.99 |
| suitcase | 0.86 | 0.9 | 0.86 | 0.61 | 0.85 | 0.91 | 0.8 | 0.77 | 0.98 | 0.92 | 0.63 | 0.8 | | 0.8 | 0.82 | 0.86 | 0.75 | 0.91 | | | 0.72 | 0.73 | 0.86 | 0.76 | 0.79 |
| teddy bear | 0.93 | 0.7 | 0.89 | 0.99 | 0.86 | 0.84 | 0.8 | 0.87 | 0.84 | 0.78 | 0.95 | 0.98 | 0.91 | 0.91 | 0.82 | 0.67 | 0.86 | 0.85 | 0.91 | 0.72 | | 0.72 | 0.91 | 0.81 | 0.75 |
| toaster | 0.63 | 0.83 | 0.67 | 0.81 | 0.86 | 0.89 | 0.9 | 0.82 | 0.82 | 0.82 | 0.65 | 0.85 | 0.88 | 0.78 | 0.71 | | 0.88 | 0.99 | 0.73 | 0.72 | | | 1 | 0.91 | 0.86 |
| truck | 0.78 | 0.81 | 0.88 | 0.84 | 0.83 | 0.82 | 0.85 | 0.69 | 0.94 | 0.95 | 0.9 | 0.89 | 0.99 | 0.89 | 0.85 | 0.86 | 0.86 | 0.72 | 0.93 | 0.86 | 0.91 | | | 0.72 | 0.96 |
| umbrella | 0.86 | 0.9 | 0.86 | 0.67 | 0.7 | 0.84 | 0.83 | 0.92 | 0.86 | 0.85 | 0.92 | 0.81 | 0.99 | 0.89 | 0.88 | 0.75 | 0.86 | 0.95 | 0.74 | 0.76 | 0.81 | 0.91 | 0.72 | | 0.78 |
| vase | 0.9 | 0.99 | 0.96 | 0.82 | 0.77 | 0.94 | 0.82 | 0.73 | 0.9 | 0.89 | 0.86 | 0.82 | 0.95 | 0.83 | 0.9 | 0.87 | 0.88 | 0.88 | 0.99 | 0.79 | 0.75 | 0.86 | 0.96 | 0.78 | |

Table 14: Per-prompt AUROC for CLIP on the counting set.

| | airplane | apple | automobile | bird | book | cake | carrot | cat | chair | cup | deer | dog | fork | frog | horse | laptop | microwave | orange | ship | suitcase | teddy bear | toaster | truck | umbrella | vase |
|---|---|---|---|---|---|---|---|---|---|---|---|---|---|---|---|---|---|---|---|---|---|---|---|---|---|
| one | 0.4 | 0.88 | 0 | | 0.48 | 1 | 0.72 | 0.08 | 0.6 | 0.5 | | 0.44 | 0.84 | | 0.49 | 0.63 | 0.5 | 0.7 | 0.27 | 0.53 | 0.92 | 0.87 | 0.19 | 0.72 | 0.57 |
| two | 0.64 | 0.81 | 0.64 | 0.31 | 0.68 | 0.63 | 0.73 | 0.84 | 0.64 | 0.61 | 0.76 | 0.65 | 0.71 | 0.39 | 0.72 | 0.69 | 0.75 | 0.79 | 0.81 | 0.75 | 0.85 | 0.76 | 0.58 | 0.59 | 0.79 |
| three | 0.76 | 0.51 | 0.62 | 0.87 | 0.58 | 0.55 | 0.52 | 0.74 | 0.52 | 0.22 | 0.6 | 0.71 | 0.67 | 0.68 | 0.64 | 0.82 | 0.59 | 0.64 | 0.68 | 0.38 | 0.74 | 0.65 | 0.78 | 0.73 | 0.31 |
| four | 0.96 | 0.57 | 0.55 | 0.7 | 0.33 | 0.44 | 0.61 | 0.52 | 0.42 | 0.42 | 0.46 | 0.65 | 0.67 | 0.65 | 0.66 | 0.73 | 0.55 | 0.58 | 0.63 | 0.54 | 0.65 | 0.58 | 0.54 | 0.72 | 0.34 |
| five | 0.6 | 0.63 | 0.6 | 0.58 | 0.67 | 0.49 | 0.56 | 0.49 | 0.61 | 0.69 | 0.63 | 0.34 | 0.39 | 0.6 | 0.44 | 0.7 | | 0.62 | 0.54 | 0.5 | 0.53 | 0.34 | 0.46 | 0.75 | 0.43 |
| six | 0.91 | 0.56 | 0.64 | 0.52 | 0.39 | 0.72 | 0.65 | 0.41 | 0.8 | 0.33 | 0.2 | 0.4 | 0.69 | 0.49 | 0.38 | 0.84 | 0.69 | 0.59 | 0.48 | 0.71 | 0.41 | 0.61 | 0.68 | 0.43 | 0.73 |

Table 15: Per-prompt AUROC for BLIP-2 on the counting set.

| | airplane | apple | automobile | bird | book | cake | carrot | cat | chair | cup | deer | dog | fork | frog | horse | laptop | microwave | orange | ship | suitcase | teddy bear | toaster | truck | umbrella | vase |
|---|---|---|---|---|---|---|---|---|---|---|---|---|---|---|---|---|---|---|---|---|---|---|---|---|---|
| one | 0.66 | 0.8 | 0.08 | | 0.38 | 0.98 | 0.79 | 0.12 | 0.35 | 0.56 | | 0.76 | 0.74 | | 0.41 | 0.48 | 0.41 | 0.75 | 0.67 | 0.78 | 0.31 | 1 | 0.9 | 0.83 | 0.61 |
| two | 0.69 | 0.88 | 0.46 | 0.4 | 0.6 | 0.79 | 0.88 | 0.12 | 0.5 | 0.57 | 0.84 | 0.6 | 0.67 | 0.72 | 0.46 | 0.6 | 0.75 | 0.88 | 0.77 | 0.94 | 0.36 | 0.77 | 0.72 | 0.86 | 0.76 |
| three | 0.6 | 0.64 | 0.54 | 0.59 | 0.79 | 0.64 | 0.58 | 0.58 | 0.41 | 0.32 | 0.48 | 0.38 | 0.71 | 0.75 | 0.6 | 0.56 | 0.5 | 0.73 | 0.5 | 0.51 | 0.49 | 0.44 | 0.67 | 0.69 | 0.41 |
| four | 0.73 | 0.59 | 0.64 | 0.5 | 0.69 | 0.4 | 0.51 | 0.28 | 0.36 | 0.4 | 0.61 | 0.45 | 0.6 | 0.58 | 0.49 | 0.52 | 0.44 | 0.65 | 0.52 | 0.51 | 0.61 | 0.52 | 0.6 | 0.81 | 0.49 |
| five | 0.3 | 0.59 | 0.63 | 0.58 | 1 | 0.56 | 0.53 | 0.52 | 0.55 | 0.46 | 0.33 | 0.27 | 0.42 | 0.55 | 0.44 | 0.67 | | 0.68 | 0.5 | 0.4 | 0.42 | 0.6 | 0.34 | 0.49 | 0.56 |
| six | 0.27 | 0.5 | 0.57 | 0.54 | 0.73 | 0.52 | 0.64 | 0.38 | 0.67 | 0.4 | 0.2 | 0.31 | 0.78 | 0.32 | 0.32 | 0.64 | 0.39 | 0.53 | 0.47 | 0.48 | 0.44 | 0.47 | 0.42 | 0.55 | 0.82 |

Table 16: Per-prompt AUROC for ImageReward on the counting set.

| | airplane | apple | automobile | bird | book | cake | carrot | cat | chair | cup | deer | dog | fork | frog | horse | laptop | microwave | orange | ship | suitcase | teddy bear | toaster | truck | umbrella | vase |
|---|---|---|---|---|---|---|---|---|---|---|---|---|---|---|---|---|---|---|---|---|---|---|---|---|---|
| one | 0.83 | 0.62 | 0.94 | | 0.81 | 1 | 0.98 | 0.73 | 0.68 | 0.46 | | 1 | 0.95 | | 0.74 | 0.59 | 0.87 | 0.75 | 0.86 | 0.82 | 0.76 | 0.99 | 1 | 0.95 | 0.59 |
| two | 0.74 | 0.97 | 0.78 | 0.66 | 0.81 | 0.8 | 0.79 | 0.96 | 0.62 | 0.89 | 0.9 | 0.9 | 0.58 | 0.72 | 0.81 | 0.82 | 0.86 | 0.92 | 0.9 | 0.96 | 0.82 | 0.94 | 0.9 | 0.9 | |
| three | 0.56 | 0.82 | 0.64 | 0.88 | 0.68 | 0.71 | 0.8 | 0.68 | 0.56 | 0.43 | 0.62 | 0.57 | 0.92 | 0.74 | 0.69 | 0.86 | 0.61 | 0.78 | 0.46 | 0.48 | 0.82 | 0.68 | 0.79 | 0.73 | 0.86 |
| four | 0.22 | 0.56 | 0.27 | 0.54 | 0.71 | 0.54 | 0.62 | 0.36 | 0.48 | 0.42 | 0.61 | 0.47 | 0.62 | 0.62 | 0.56 | 0.6 | 0.52 | 0.69 | 0.62 | 0.59 | 0.4 | 0.41 | 0.61 | 0.6 | 0.47 |
| five | 0.69 | 0.76 | 0.47 | 0.45 | 0.95 | 0.59 | 0.58 | 0.46 | 0.37 | 0.4 | 0.52 | 0.45 | 0.79 | 0.45 | 0.64 | 0.66 | | 0.67 | 0.71 | 0.47 | 0.45 | 0.38 | 0.68 | 0.76 | 0.5 |
| six | 0.14 | 0.58 | 0.39 | 0.31 | 0.92 | 0.63 | 0.62 | 0.47 | 0.3 | 0.39 | 0.23 | 0.33 | 0.82 | 0.43 | 0.46 | 0.71 | 0.54 | 0.48 | 0.53 | 0.55 | 0.42 | 0.54 | 0.61 | 0.53 | 0.81 |

Table 17: Per-prompt AUROC for PickScore on the counting set.

| | airplane | apple | automobile | bird | book | cake | carrot | cat | chair | cup | deer | dog | fork | frog | horse | laptop | microwave | orange | ship | suitcase | teddy bear | toaster | truck | umbrella | vase |
|---|---|---|---|---|---|---|---|---|---|---|---|---|---|---|---|---|---|---|---|---|---|---|---|---|---|
| one | 0.78 | 0.91 | 1 | | 0.77 | 1 | 0.82 | 0.18 | 0.57 | 0.67 | | 1 | 0.85 | | 0.78 | 0.63 | 0.88 | 0.91 | 0.72 | 0.9 | 0.94 | 0.88 | 0.95 | | 0.59 |
| two | 0.85 | 0.9 | 0.86 | 0.91 | 0.79 | 0.78 | 0.8 | 1 | 0.96 | 0.68 | 0.98 | 0.63 | 0.89 | 0.94 | 0.74 | 0.87 | 0.82 | 0.93 | 0.95 | 0.94 | 0.99 | 0.8 | 0.92 | 0.82 | 0.93 |
| three | 0.81 | 0.9 | 0.79 | 0.87 | 0.65 | 0.69 | 0.94 | 0.84 | 0.77 | 0.7 | 0.82 | 0.86 | 0.93 | 0.8 | 0.76 | 0.78 | 0.8 | 0.89 | 0.7 | 0.68 | 0.88 | 0.8 | 0.73 | 0.93 | 0.89 |
| four | 0.78 | 0.82 | 0.57 | 0.82 | 0.81 | 0.83 | 0.76 | 0.58 | 0.59 | 0.7 | 0.7 | 0.76 | 0.81 | 0.77 | 0.65 | 0.75 | 0.88 | 0.8 | 0.56 | 0.82 | 0.53 | 0.75 | 0.74 | 0.82 | 0.71 |
| five | 0.62 | 0.58 | 0.42 | 0.43 | 0.5 | 0.58 | 0.65 | 0.38 | 0.49 | 0.43 | 0.85 | 0.3 | 0.86 | 0.33 | 0.78 | 0.66 | | 0.75 | 0.65 | 0.51 | 0.51 | 0.43 | 0.67 | 0.73 | 0.76 |
| six | 0.33 | 0.69 | 0.36 | 0.54 | 0.88 | 0.54 | 0.65 | 0.55 | 0.18 | 0.43 | 0.68 | 0.34 | 0.71 | 0.61 | 0.56 | 0.66 | 0.46 | 0.53 | 0.83 | 0.59 | 0.54 | 0.72 | 0.63 | 0.87 | 0.89 |

Table 18: Per-prompt AUROC for TextNorm on the counting set.

| | airplane | apple | automobile | bird | book | cake | carrot | cat | chair | cup | deer | dog | fork | frog | horse | laptop | microwave | orange | ship | suitcase | teddy bear | toaster | truck | umbrella | vase |
|---|---|---|---|---|---|---|---|---|---|---|---|---|---|---|---|---|---|---|---|---|---|---|---|---|---|
| one | 0.75 | 0.7 | 0.08 | | 0.86 | 1 | 1 | 0.24 | 0.72 | 0.46 | | 0.96 | 0.97 | | 0.67 | 0.5 | 0.63 | 0.86 | 0.43 | 0.99 | 0.82 | 0.9 | 0.96 | 0.82 | 0.57 |
| two | 0.77 | 0.96 | 0.8 | 0.81 | 0.91 | 0.93 | 0.94 | 1 | 1 | 0.64 | 0.93 | 0.49 | 0.9 | 0.95 | 0.7 | 0.65 | 0.83 | 0.95 | 0.93 | 0.98 | 0.96 | 0.93 | 0.87 | 0.79 | 0.91 |
| three | 0.87 | 0.95 | 0.85 | 0.94 | 0.7 | 0.96 | 0.96 | 0.85 | 0.88 | 0.82 | 0.81 | 0.9 | 0.88 | 0.9 | 0.82 | 0.65 | 0.94 | 0.98 | 0.75 | 0.8 | 0.95 | 0.93 | 0.78 | 0.8 | 0.88 |
| four | 0.7 | 0.88 | 0.86 | 0.84 | 0.95 | 0.87 | 0.93 | 0.67 | 0.72 | 0.88 | 0.8 | 0.93 | 0.93 | 0.82 | 0.66 | 0.74 | 0.98 | 0.89 | 0.9 | 0.92 | 0.67 | 0.9 | 0.84 | 0.9 | 0.77 |
| five | 0.9 | 0.69 | 0.72 | 0.73 | 0.56 | 0.65 | 0.86 | 0.74 | 0.93 | 0.78 | 0.9 | 0.61 | 0.81 | 0.83 | 0.98 | 0.84 | | 0.9 | 0.69 | 0.73 | 0.86 | 0.73 | 0.69 | 0.66 | 0.92 |
| six | 0.98 | 0.83 | 0.64 | 0.76 | 0.82 | 0.87 | 0.74 | 0.39 | 0.75 | 0.65 | 0.91 | 0.66 | 0.88 | 0.78 | 0.78 | 0.86 | 0.54 | 0.68 | 0.89 | 0.77 | 0.82 | 0.91 | 0.59 | 0.82 | 0.78 |

### B.4 REWARD MODEL SCORES ON SAMPLE IMAGES

While PickScore generally outperforms the other baselines, there are instances where those baselines exhibit stronger alignment as shown in Figure 8. This observation naturally led to considering ensemble methods to achieve further improvement.

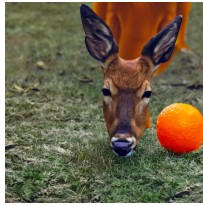

| Prompt | CLIP | BLIP-2 | IR | PS |
|---|---|---|---|---|
| a deer and an orange | **0.385** | **0.407** | **1.750** | 0.232 |
| a deer | 0.314 | 0.347 | -0.028 | 0.207 |
| a deer and two oranges | 0.368 | 0.391 | 0.867 | **0.236** |
| a deer-like orange | 0.344 | 0.394 | 1.454 | 0.221 |
| an orange-like deer | 0.348 | 0.391 | 1.023 | 0.222 |

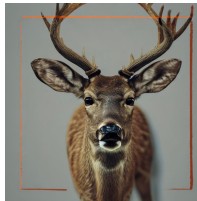

| Prompt | CLIP | BLIP-2 | IR | PS |
|---|---|---|---|---|
| a deer and an orange | 0.301 | 0.411 | 0.377 | **0.228** |
| a deer | **0.314** | **0.432** | **1.097** | 0.220 |
| a deer and two oranges | 0.295 | 0.388 | -0.906 | 0.222 |
| a deer-like orange | 0.304 | 0.372 | 0.125 | 0.222 |
| an orange-like deer | 0.321 | 0.394 | 0.225 | 0.224 |

Figure 8: Reward model scores on sample images and prompts.

## C MORE HUMAN EVALUATION RESULTS

For the best-of-$n$ sampling experiment, annotators evaluate the top four images selected from the $n$ samples based on the reward models for each prompt. As illustrated by the evaluation results summarized in Figure 9, TextNorm consistently outperforms all baselines in terms of text-image alignment, often by a substantial margin, across all three values of $n$. Especially compared to CLIP and BLIP-2, TextNorm achieves notably better text-image alignment as well as image quality. Even when compared to ImageReward and PickScore, TextNorm achieves two to three times as many wins as losses in alignment, with comparable or only minor compromises in image quality.

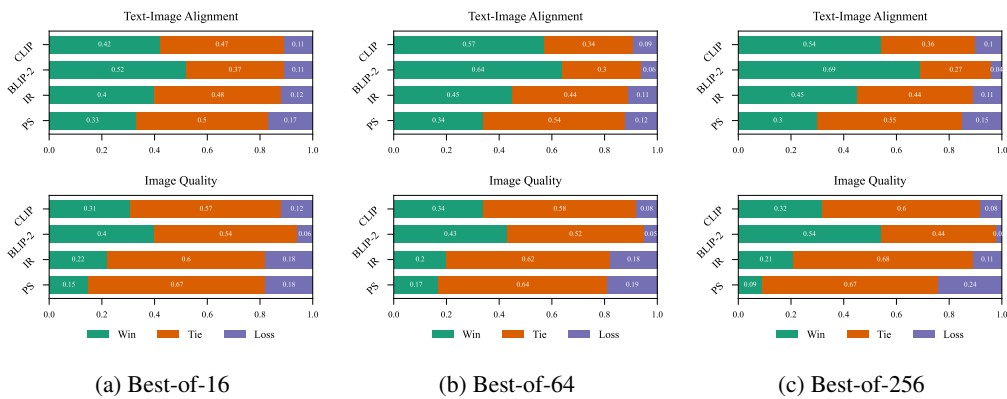

Figure 9: Results of human evaluation of the images selected using best-of-$n$ sampling.

# D DETAILS ON PROMPT SYNTHESIS VIA LLMS

Algorithm 1 outlines detailed pseudocode for instructing ChatGPT to generate a contrastive prompt set based on a given input prompt $x_0$. Depending on the nature of $x_0$, we assign it appropriate categories and provide an optional list of few-shot examples to guide generation through in-context learning. For example, given the text prompt *"A black colored car"*, we categorize it under *colors* and *counting* and provide few-shot examples that encompass common failure cases for the two categories. In our experiments, we use the gpt-4-0613 model for contrastive prompt generation, configuring the temperature to 0.0 and the frequency penalty to 0.2. These hyperparameters affect the generation process based on the token frequencies.

Table 19 presents the few-shot examples considered in our experiments for eight distinct categories: *text*, *style*, *composition*, *counting*, *creative*, *location*, *colors*, and *spatial*. To generate diverse and high-quality prompts, we vary the quantity and type of objects described in the input prompt, along with other relevant properties like spatial relations.

---

**Algorithm 1** Prompt set synthesis via ChatGPT

---

```
# x_0: input prompt
# T, P: temperature, frequency penalty
# examples_per_category: few-shot examples for each category

# invoke ChatGPT to return chat completion
def model_completion(x_0, model_prompt, **params):
    model_prompt = model_prompt.copy()
    model_prompt.append({"role": "user", "content": x_0})
    resp = openai.ChatCompletion.create(
        engine="gpt-4-0613",
        messages=model_prompt,
        **params)
    return resp

# prepare the LLM prompt with few-shot examples and invoke the model
def synthesize_prompts(x_0, T, P):
    messages = [{
        "role": "system",
        "content": "Create captions that are different from the original
            input used for the text-to-image generation model,
            referencing the provided failure cases. The new captions
            should offer perspectives that are distinct from the original
             context of the images. Ensure that each contrasting caption
            provides a distinct perspective, while maintaining the
            integrity of the image's subject matters. Let's think step by
             step."
    }]
    # allocate category to input prompt x_0
    categories = category_allocation(x_0)
    for category in categories:
        few_shot_examples = category + "[\n"
        for example in examples_per_category[category]:
            few_shot_examples += "Original prompt:" + example["
                original_prompt"]+"\n"
            few_shot_examples += "Contrasting captions:"
            for idx, caption in enumerate(example["proper_candidates"]):
                few_shot_examples += f"{idx+1}.{caption}\n"
        few_shot_examples += "]\n"
    messages[0]["content"] += few_shot_examples
    params = {"temperature": T, "frequency_penalty": P}
    return model_completion(x_0, messages, params)
```

---

Table 19: Few-shot examples for each category of prompts used for in-context learning.

**Category:** Text

**Input:** "A sign that says 'Diffusion'."

**Output:** ["A sign misspelled as 'Difision'.", "A sign containing a bizarre accent 'Difśion'."]

**Category:** Style

**Input:** "Greek statue of a man tripping over a cat."

**Output:** ["Greek statue of a man","Greek statue of two men","Greek statue of two men tripping over a cat.","Greek statue of a man tripping over a dog.","Greek statue of two men tripping over a dog.","Greek statue of a man tripping over a ball."]

**Category:** Composition

**Input:** "A red car and a white sheep."

**Output:** ["A red car and two white sheep.","A red car and a herd of sheep.","A red car and a dominant white sheep among the gray ones.","A red car with two real sheep on the side."]

**Category:** Counting

**Input:** "Four cars on the street."

**Output:** ["Two cars on the street.","Three cars on the street.","Five cars on the street.","Six cars on the street."]

**Category:** Creative

**Input:** "A heart made of chocolate"

**Output:** ["a star made of caramel","a flower made of marshmallows,","a diamond made of gummy bears","a moon made of licorice","a sun made of jelly beans","a butterfly made of lollipops","a crown made of cotton candy","a rainbow made of skittles","a cloud made of cotton candy","a tree made of chocolate-covered pretzels","a snowflake made of peppermint candies","a fish made of sour gummies","a bird made of chocolate-covered almonds","a car made of chocolate bars","a house made of chocolate cookies","a boat made of chocolate-covered strawberries","an airplane made of chocolate truffles","a guitar made of chocolate wafer sticks","a camera made of chocolate coins","a dinosaur made of chocolate eggs"]

**Category:** Location

**Input:** "A glowing mushroom in the forest"

**Output:** ["a sparkling flower in the garden","a luminous firefly in the night sky","a shimmering starfish in the ocean","a radiant sunflower in the field","a glowing jellyfish in the deep sea","a gleaming diamond in the jewelry store","a luminescent moon in the night sky","a glowing firefly in the meadow","a sparkling gemstone in the cave","a luminous butterfly in the garden","a shimmering seashell on the beach","a radiant rainbow in the sky","a glowing lantern in the dark","a luminescent lightning bug in the field","a sparkling crystal in the cave","a shimmering waterfall in the forest","a radiant star in the night sky","a glowing firefly in the park","a luminous pearl in the oyster","a sparkling diamond in the jewelry box"]

**Category:** Colors

**Input:** "A blue bird and a brown bear."

**Output:** ["A pair of bears, one blue and the other brown.","A blue bear and a brown bear, no bird in sight.","Two bears, both brown, no blue bird.","Two bears, one brown and the other unexpectedly blue."]

**Category:** Spatial

**Input:** "An umbrella on top of a spoon."

**Output:** ["A spoon.","An umbrella.","An umbrella on the right of a spoon.","An umbrella on the left of a spoon.","An umbrella at the bottom of a spoon.","Two umbrellas on top of a spoon."]

