# OpenReview forum: "Confidence-aware Reward Optimization for Fine-tuning Text-to-Image Models"
_ICLR.cc/2024/Conference — ICLR 2024 poster_

### Official Review · Reviewer_fRcb · 2023-10-30

**Soundness:** 3 good
**Presentation:** 3 good
**Contribution:** 3 good
**Rating:** 6
**Confidence:** 4

**Summary:**

This paper study the problem of reward functions on human feedback data for image diffusion models. It proposes a new text-image alignment assessment benchmark (TIA2) for evaluating the problem of reward designing. It shows that several existing reward-based models are not well-aligned with human assessment after evaluated on the benchmark. The authors propose a method TextNorm to induce confidence calibration in reward models. It shows the proposed score help reduce over-optimization in the human feedback tuning. Experiments are conducted with Stable Diffusion v2.1 and LoRA for parameter-efficient fine-tuning.

**Strengths:**

-	The proposes benchmark for evaluating text-image alignment can help the reward designing in human feedback data. It includes different categories of prompts: comprehensive, counting and composition, which are the key categories that common image reward functions such as CLIP and ImageReward cannot do well on.
-	The proposed TextNorm is novel and shows superior performance on improving calibration errors on different types of prompts. It also improves text-image alignment compared to existing reward functions according to some qualitative results and human evaluation.
-	An ablation study is also conducted to analyze the impact of different aspects: normalization over prompts, prompt synthesis, and the textual inconsistency score.

**Weaknesses:**

-	In the process of prompt synthesis using language model, the authors only ask the LLM to generate variations based on the object category and number of objects. Other important aspects such as spatial relationships and object attributes like material, color, size is not included. This greatly limit the scope of the experiment in this work. I also only saw sampled result images related to object category and object number in this paper. It would be great if more analysis on those other aspects can be studied as well.
-	In figure 5, the difference of TextNorm and existing methods are not significant. BLIP2 has very similar results as the TextNorm on the correctness of the object number and categories.

**Questions:**

In the human evaluation, the authors are comparing with “the best-performing baseline reward model” but did not mention which one it is. Please provide the details on which reward function you are comparing with.

---

> ### Author Response · Authors · 2023-11-17
>
> Dear Reviewer fRcb,
>
> We sincerely appreciate your review with thoughtful comments. We have carefully considered each of your questions and provide detailed responses below. Please let us know if you have any further questions or concerns.
>
> ---
>
> **[W1] Additional aspects of text-image alignment in prompt synthesis.**
>
> We instruct the LLM to generate variations with an emphasis on the categories and quantities of objects, as they are common alignment issues with text-to-image models. However, we also utilize different sets of few-shot examples tailored to the type of input prompt to guide the LLM towards generating relevant variations. As an example, for the prompt “a red box on top of a blue box” in Table 14, the set of synthesized prompts includes variations such as “two blue balls on top of a pink box” and “a black triangle on top of a green box”, exploring alternative color attributes for the objects. We have revised the draft to add these details in Section 4.1.
>
> ---
>
> **[W2] Qualitatively comparable results.**
>
> Note that Figures 1 and 4 contain examples that further highlight the shortcomings of the baseline models, including BLIP-2. We also added Figure 8 with an additional example that better showcases the limitations of BLIP-2 in the revised draft. Given the variability in performance across prompts, we remark that it is important to supplement the qualitative results with a human evaluation, as we have done in the work.
>
> ---
>
> **[Q1] Details on the baseline models in human eval.**
>
> Thank you for pointing this out. To clarify, PickScore is used as the baseline for the composition and counting prompts, while ImageReward is the baseline for the comprehensive prompts. We have also revised our draft to include the details.

---

> > ### Author Response · Authors · 2023-11-21
> > **Gentle reminder**
> >
> > Dear Reviewer fRcb,
> >
> > We once again thank you for your review of our manuscript. We sincerely appreciate your valuable insights and feedback.
> >
> > As the discussion period is drawing to a close, we kindly remind you that there are two days remaining for further comments or questions. We would be grateful for the opportunity to address any additional concerns you may have before the discussion phase ends.
> >
> > Thank you for your time and consideration.
> >
> > Sincerely,
> >
> > Authors

---

> > ### Comment · Reviewer_fRcb · 2023-11-22
> >
> > Thanks for addressing my comments. It is good to know that this work also include attributes into the prompt synthesis. However, other important factors are still missing: spatial relationship, object sizes etc. I would like to see these factors as well but the current version seems to have sufficient contribution. With all that considerations, I still hold my original rating.

---

### Official Review · Reviewer_exWV · 2023-11-01

**Soundness:** 3 good
**Presentation:** 3 good
**Contribution:** 2 fair
**Rating:** 6
**Confidence:** 4

**Summary:**

This paper proposed a text-image alignment benchmark and reward score normalization methods to evaluate and address the "overoptimization" issue when finetuning with a reward model. The proposed benchmark contains 550 prompts covering comprehensive/counting/compostion categories and 27500 generated images. The normalization method consider normalizing text-image reward scores by contrasting to generated texts. The human evaluation and qualitative results showing the proposed method performs better on the "overoptimization" issue over other baselines.

**Strengths:**

- The paper is well-written and easy to understand
- The proposed benchmark showing the drawbacks of different reward models
- The proposed text normalization method can reduce the overoptimization problem and improves the model finetuning performance

**Weaknesses:**

- One major contribution is the proposed text-image alignment assessment benchmark. However, the role/completeness of the benchmark are not fully explored. For example: 1. Are the comprehensive/counting/composition enough to compare those reward models? 2. In figure 3 (a), we can see ImageReward > PickScore > CLIP > BLIP-2 in terms of "comprehensive". Are these reward model ranking align with human evaluation (Do human also thinks ImageReward > PickScore > CLIP > BLIP-2)?

**Questions:**

- Section 5.2. "select the top 10% of the samples as measured by the chosen reward model". Is that means the model finetuned with different reward models are finetuned with different data? Why not use the same data for finetuning?
- Any Failure cases for proposed text norm method? (failure cases for both comprehensive/counting/composition are helpful)

---

> ### Author Response · Authors · 2023-11-17
>
> Dear Reviewer exWV,
>
> We sincerely appreciate your review with thoughtful comments. We have carefully considered each of your questions and provide detailed responses below. Please let us know if you have any further questions or concerns.
>
> ---
>
> **[W1] Are the prompts in the benchmark sufficient for comparing the reward models?**
>
> We believe our benchmark offers a sufficiently diverse collection of prompts for evaluating reward models across various aspects of text-image alignment. We introduced separate composition and counting sets to specifically assess the ability of the reward models to evaluate compositional generation, a known challenge for text-to-image models, particularly when involving multiple objects [1]. To increase prompt diversity, we added the comprehensive set, which contains 100 prompts spanning nine fine-grained semantic categories: creative, spatial, text rendering, style, counting, color, composition, location, and open-ended. As a future direction, we aim to expand the scope of our benchmark by incorporating additional prompts, images, and human labels.
>
> ---
>
> **[W2] Do the rankings in Figure 3 align with human assessments?**
>
> Figure 3 indeed summarizes the overall alignment between the reward models and human assessments, as measured by the expected calibration error (ECE) computed with the human labels from the benchmark. Therefore, it is reasonable to conclude that a reward model for which more of the prompts fall within lower ECE ranges is overall more aligned with human preferences.
>
> ---
>
> **[Q1] Are the models fine-tuned with different data in Section 5.2?**
>
> The same offline dataset of images was considered for all fine-tuned models. However, filtered samples for each reward model may vary depending on how the model evaluates them. This training approach was used for two reasons: (a) using synthetic images from generative models filtered for quality is a common practice for downstream task training [2,3,4], and (b) we empirically found fine-tuning on filtered samples to be significantly more effective than using the entire dataset, which may contain low-quality samples, for optimizing for the given reward.
>
> ---
>
> **[Q2] Failure cases of TextNorm**
>
> Our method employs a set of semantically contrastive prompts to enhance calibration by normalizing reward scores over these prompts. In case the set contains semantically equivalent or irrelevant prompts, normalization can result in worse calibration. Below are the results from an ablation experiment where we compare semantically contrastive prompts to random prompts in terms of calibration. This experiment underscores the importance of prompt set construction, highlighting that a poorly chosen set could yield worse results.
>
> | Reward |  | Comprehensive |  |  | Counting |  |  | Composition |  |
> |---|---|:---:|---|---|:---:|---|---|:---:|---|
> |  | ECE | ACE | MCE | ECE | ACE | MCE | ECE | ACE | MCE |
> | Random | 0.2777 | 0.2094 | 0.4986 | 0.3071 | 0.2323 | 0.5565 | 0.1758 | 0.1364 | 0.3883 |
> | LLM | 0.2516 | 0.1809 | 0.4852 | 0.2213 | 0.1609 | 0.4635 | 0.1604 | 0.1146 | 0.3666 |
>
> ---
>
> **References**
>
> [1] Feng, Weixi, et al. “Training-Free Structured Diffusion Guidance for Compositional Text-to-Image Synthesis.” ICLR 2023.
>
> [2] He, Ruifei, et al. “Is synthetic data from generative models ready for image recognition?” ICLR 2023.
>
> [3] Fan, Ying, et al. “DPOK: Reinforcement Learning for Fine-tuning Text-to-Image Diffusion Models.” arXiv 2023.
>
> [4] Schuhmann, Christoph, et al. “LAION-400M: Open Dataset of CLIP-Filtered 400 Million Image-Text Pairs.” arXiv 2021.

---

> > ### Author Response · Authors · 2023-11-21
> > **Gentle reminder**
> >
> > Dear Reviewer exWV,
> >
> > We once again thank you for your review of our manuscript. We sincerely appreciate your valuable insights and feedback.
> >
> > As the discussion period is drawing to a close, we kindly remind you that there are two days remaining for further comments or questions. We would be grateful for the opportunity to address any additional concerns you may have before the discussion phase ends.
> >
> > Thank you for your time and consideration.
> >
> > Sincerely,
> >
> > Authors

---

### Official Review · Reviewer_9bKW · 2023-11-01

**Soundness:** 3 good
**Presentation:** 2 fair
**Contribution:** 3 good
**Rating:** 6
**Confidence:** 3

**Summary:**

The paper proposes a simple method to prevent the overoptimization problem in alignment of text-to-image models. They work hypothesizes that the overoptimization problem is mainly due to lack of alignment between the human preference and the learnt reward model. Inorder to solve this issue, they propose two techniques namely textual normalization which better normalizes the reward using contrastive text description. Further they propose textual inconsistency score, which compares the distance between the textual description and the predicted reward . They find that both these objectives help improve the calliberation and alignment with the human reward.

**Strengths:**

- The problem that paper picks namely reward overoptimization. Seems to be important with very works addressing it.
- The paper does a good job at benchmarking previous reward functions on various settings.
- The paper proposes novel techniques to improve the alignment and caliberation of the current reward models.

**Weaknesses:**

- The paper talks about overoptimization as the motivation, however doesn't directly evaluate for overoptimization. It instead evaluates for caliberation and human evaluation. To my understanding , current works get away with overoptimization issue with early stopping such as AlignProp (https://arxiv.org/abs/2310.03739) however this is not discussed in the paper. I would expect the right way to evaluate would be to not do early stopping and see the tradeoffs with the proposed solutions. Also talk about the benefits of not doing early stopping.
- Further AlignProp, says that they only need to do early stopping for the Aesthetics reward function, however not for HPS v2. I think it would be worth comparing againt the HPS v2 reward function and also discuss methods such as AlignProp or DDPO (https://arxiv.org/abs/2305.13301) and the tricks they use to prevent overoptimization. And why using those tricks might not be a good idea.

**Questions:**

- Is there a fundamental reason why Softmax CLIP objective is better at caliberation than Sigmoid CLIP objective(https://arxiv.org/abs/2303.15343) ?

-  "Hence, we use the insight that the terms that contribute significantly to the denominator of the softmax score are the ones for which rφ(x, y) is sufficiently large. " -  It's unclear to me how using semantic-contrastive semantic prompts results in finding the terms that have high reward value?

- Can the authors instead of finding contrastive prompts, take random prompts and show an ablation?

---

> ### Author Response · Authors · 2023-11-17
>
> Dear Reviewer 9bKW,
>
> We sincerely appreciate your review with thoughtful comments. We have carefully considered each of your questions and provide detailed responses below. Please let us know if you have any further questions or concerns.
>
> ---
>
> **[W1] Evaluating overoptimization**
>
> In our study, we consider human assessment of text-image alignment as the true objective and base our investigation on human annotations and evaluations. For our fine-tuning experiments, we evaluate the reward models against this true objective by conducting a human evaluation (the true objective) and report the results in Figure 6. The calibration analysis, which is also based on the human annotations from our benchmark, is for understanding the limitations of existing rewards and the associated risk of overoptimization.
>
> ---
>
> **[W2] Connection to early stopping**
>
> Thank you for your pointer. While early stopping is another viable approach (orthogonal to ours) to addressing overoptimization, it has several limitations. One challenge is determining the optimal stopping point, especially when evaluating the true objective is expensive. As you pointed out, the authors of AlignProp use early stopping in their experiments to guard against “loss of image-text alignment if optimized above a specific score” [1]. However, it is difficult to justify whether the 10th epoch is the optimal point at which to stop training, without conducting many human evaluations on multiple checkpoints. The authors of DDPO [2] recognize that overoptimization is an issue in text-to-image generation but do not discuss any specific methods for addressing it.
>
> We believe that our method of enhancing reward model calibration through alignment with human assessment can be also integrated into early stopping to more accurately determine when to halt optimizing against proxy rewards. We have revised our draft to include a discussion comparing early stopping and our proposed method.
>
> ---
>
> **[W3] Evaluation of the HPS v2 reward model.**
>
> We evaluated HPS v2 on our benchmark based on calibration metrics and include the results in the table below.
>
> | Reward |  | Comprehensive |  |  | Counting |  |  | Composition |  |
> |---|---|:---:|---|---|:---:|---|---|:---:|---|
> |  | ECE | ACE | MCE | ECE | ACE | MCE | ECE | ACE | MCE |
> | HPS v2 | 0.2801 | 0.2007 | 0.5581 | 0.2579 | 0.1881 | 0.5064 | 0.2266 | 0.1653 | 0.4718 |
> | TextNorm | 0.2516 | 0.1809 | 0.4852 | 0.2213 | 0.1609 | 0.4635 | 0.1604 | 0.1146 | 0.3666 |
>
> HPS v2 is indeed a competitive baseline compared to the other reward models we consider in this work. However, TextNorm still outperforms HPS v2 on our benchmark. We have revised our draft to include the results in Appendix.
>
> ---
>
> **[Q1] Softmax CLIP objective vs. sigmoid CLIP objective.**
>
> While our proposed TextNorm (Eq. 1) has a notational similarity to the softmax-based CLIP objective, it is important to note that we are not claiming that the softmax-based objective is superior to the SigLIP objective [1] in terms of calibration. Our main focus is to evaluate the effectiveness of normalizing reward scores across contrastive prompts on improving calibration. The proposed method does not depend on the training objective of reward models. Nevertheless, we think evaluating SigLIP-based models is an interesting future question to explore.
>
> ---
>
> **[Q2] Use of semantically contrastive prompts.**
>
> The underlying rationale behind our prompt set design, which employs semantically contrastive prompts that are syntactically similar, is based on the following hypothesis: prompts $x$ that differ from the input prompt $x_0$ in both syntax and semantics are unlikely to generate high reward values $r_{\phi}(x, y)$. Random prompts serve as an example of such $x$. As we show in our response to [Q3] in this letter, the prompt set using our method outperforms a random set of prompts, supporting the validity of the hypothesis. We have clarified this point in the revised draft.

---

> > ### Author Response · Authors · 2023-11-17
> >
> > **[Q3] An ablation using random prompts.**
> >
> > Thank you for your suggestion. We conducted an additional experiment to compare semantically contrastive prompts to random prompts in terms of calibration. We used 20 randomly selected prompts from the comprehensive set to evaluate the counting and composition sets. To evaluate the comprehensive set itself, we excluded the input prompt and randomly sampled from the rest.
> >
> > | Reward |  | Comprehensive |  |  | Counting |  |  | Composition |  |
> > |---|---|:---:|---|---|:---:|---|---|:---:|---|
> > |  | ECE | ACE | MCE | ECE | ACE | MCE | ECE | ACE | MCE |
> > | Random | 0.2777 | 0.2094 | 0.4986 | 0.3071 | 0.2323 | 0.5565 | 0.1758 | 0.1364 | 0.3883 |
> > | LLM | 0.2516 | 0.1809 | 0.4852 | 0.2213 | 0.1609 | 0.4635 | 0.1604 | 0.1146 | 0.3666 |
> >
> > The experiment emphasizes the importance of using a relevant set of contrastive prompts and demonstrates that a random set of prompts can lead to inferior calibration. The results also underscore the effectiveness of the LLM-based approach to generating contrastive prompts. We have also revised the draft to add the results.
> >
> > **References**
> >
> > [1] Prabhudesai, Mihir, et al. “Aligning Text-to-Image Diffusion Models with Reward Backpropagation.” arXiv 2023.
> >
> > [2] Black, Kevin, et al. “Training Diffusion Models with Reinforcement Learning.” arXiv 2023.

---

> > > ### Author Response · Authors · 2023-11-21
> > > **Gentle reminder**
> > >
> > > Dear Reviewer 9bKW,
> > >
> > > We once again thank you for your review of our manuscript. We sincerely appreciate your valuable insights and feedback.
> > >
> > > As the discussion period is drawing to a close, we kindly remind you that there are two days remaining for further comments or questions. We would be grateful for the opportunity to address any additional concerns you may have before the discussion phase ends.
> > >
> > > Thank you for your time and consideration.
> > >
> > > Sincerely,
> > >
> > > Authors

---

### Author Response · Authors · 2023-11-17

Dear Reviewers,

We sincerely appreciate your time and effort in reviewing our manuscript. As the reviewers highlighted, our work introduces a benchmark consisting of text prompts, images and human annotations valuable for assessing reward models in text-to-image generation (9bKW, exWV, fRcb). We evaluate several state-of-the-art reward models on the benchmark and find that they are often poorly aligned with human assessments (exWV, fRcb). Based on the analysis, we propose a simple yet effective method to enhance alignment by normalizing the rewards across a set of contrastive prompts. We demonstrate its effectiveness through fine-tuning experiments accompanied by a human evaluation (9bKW, exWV, fRcb). We are also pleased that the reviewers find our manuscript well-written and easy to understand (exWV) and the additional analyses, including the ablation study, to be helpful (fRcb).

We are grateful for the reviewers’ insightful feedback on our manuscript. We have incorporated the suggestions into our revised draft, which includes additional results and revisions marked in red. Specifically, we have implemented the following changes:
- A discussion on early stopping as an alternative approach to addressing overoptimization.
- A detailed explanation on the use of few-shot examples in LLM-based prompt synthesis.
- An ablation study comparing the use of random prompt sets and LLM-generated prompts..
- Additional qualitative examples illustrating the limitations of baseline reward models.
- An evaluation of HPS v2, a recently introduced reward model trained on large human preference data, on our benchmark.

In our individual responses, we have also addressed each reviewer's specific comments and questions. Please do not hesitate to follow up with any further inquiries.

We are sincerely grateful for the reviewers’ invaluable contributions to enhancing our manuscript. We believe that the refined manuscript will more effectively deliver the significance of our work to the wider research community.

Thank you,

Authors

---

### Meta-Review · Area_Chair_v7R5 · 2023-12-17

**Metareview:**

This paper introduces a benchmark comprising diverse text prompts, images, and human annotations and evaluates state-of-the-art reward models on the benchmark to find that they are often poorly aligned with human assessments. All reviewers are overall supportive of this paper. Authors give detailed feedbacks. The ACs thus decided to accept it.

**Justification For Why Not Higher Score:**

The completeness of the benchmark is not fully explored.

**Justification For Why Not Lower Score:**

This paper  proposes a simple yet effective method for enhancing the alignment.

---

### Decision · Program_Chairs · 2024-01-16

Accept (poster)